# On the Choice of Learning Rate for Local SGD

**Lukas Balles**[‡]                                                 *lukas.balles@aleph-alpha.com*
*Aleph Alpha, Heidelberg, Germany. Work done at AWS.*

**Prabhu Teja S**[‡]                                                        *prbuteja@amazon.de*
*Amazon Web Services, Berlin, Germany.*

**Cédric Archambeau**                                              *cedric.archambeau@helsing.ai*
*Helsing, Berlin, Germany. Work done at AWS.*

**Reviewed on OpenReview:** *https://openreview.net/forum?id=DPvwr4HJdt*

## Abstract

Distributed data-parallel optimization accelerates the training of neural networks, but requires constant synchronization of gradients between the workers, which can become a bottleneck. One way to reduce communication overhead is to use Local SGD, where each worker asynchronously takes multiple local gradient steps, after which the model weights are averaged. In this work, we discuss the choice of learning rate for Local SGD, showing that it faces an intricate trade-off. Unlike in the synchronous case, its gradient estimate is biased, with the bias dependent on the learning rate itself. Thus using learning rate scaling techniques designed for faster convergence in the synchronous case with Local SGD results in a performance degradation as previously observed. To analyze the manifestation of this bias, we study convergence behaviour of Local SGD and synchronous data-parallel SGD when using their optimal learning rates. Our experiments show that the optimal learning rate for Local SGD differs substantially from that of SGD, and when using it the performance of Local SGD matches that of SGD. However, this performance comes at the cost of added training iterations, rendering Local SGD faster than SGD only when communication is much more time-consuming than computation. This suggests that Local SGD may be of limited practical utility.

## 1 Introduction

Gradient-based optimization techniques like Stochastic Gradient Descent (SGD) (Robbins & Monro, 1951) and its variants (Qian, 1999; Sutskever et al., 2013; Kingma & Ba, 2015) have contributed enormously to the success of deep learning models in the past decade. With increasing scale of both models and datasets, *distributed* training has become commonplace. The predominant distributed training paradigm is *data-parallel* training, where each of $K$ workers computes a gradient using an independently drawn minibatch of data in parallel. These individual gradients are then averaged across all workers before an optimizer update is applied, effectively increasing the batch size by a factor of $K$.

The use of larger batch sizes results in an improved gradient estimate, its variance being inversely proportional to the total batch size (Bottou et al., 2018). However, fully capitalizing on this reduced variance requires higher learning rates. Goyal et al. (2018) propose to increase the learning rate linearly with the number of workers. The goal is to achieve what is called perfect *linear scaling*, *i.e.*, to converge in $K$ times fewer iterations when using $K$ workers. This has been shown to succeed for a moderate number of workers,

---

[‡]Equal Contribution.

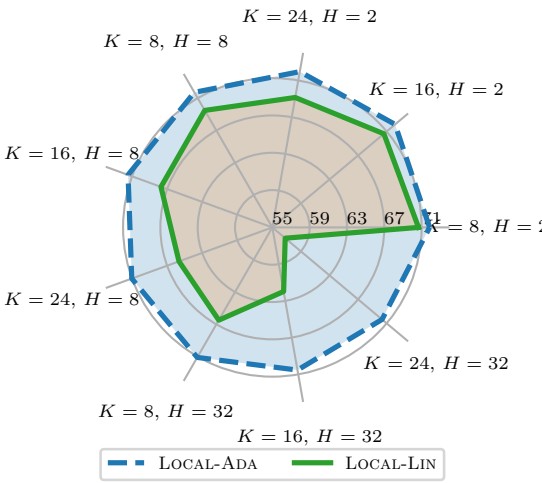

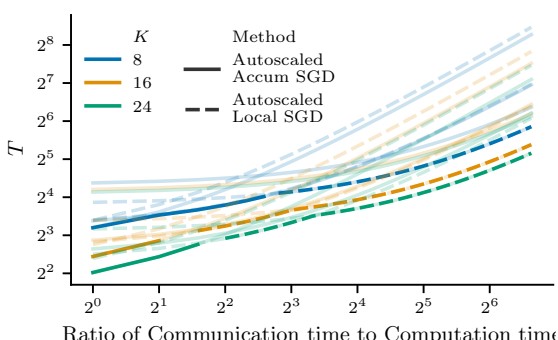

Figure 1: Why is Local SGD's performance lower than SGD? Existing experimental works on Local SGD (Lin et al., 2020) use linear scaling that was designed for synchronous SGD and thus result in poorer performance when number of local steps $H$ is large (LOCAL-LIN). When using a automatic learning rate scaling method (LOCAL-ADA) based on optimal learning rates, we find that Local SGD performs similarly for a large range of $K$ and $H$. These results are for a Wide ResNet-28 being trained on ImageNet32. Each line represents a $K, H$ configuration and accuracy increases moving away from the center.

Figure 2: When is Local SGD useful in practice? With adaptive learning rate scaling methods modulating the number of training iterations, we examine when Local SGD (LOCAL-ADA) faster than SGD (ACC-ADA). We see that Local SGD is faster than the synchronous case only when communication costs much more than computation ($m \geq 2$), thus making Local SGD of utility only in scenarios of extreme communication costs. We show on x-axis the communication to computation costs of a hypothetical system, and pseudo-wall clock time on y-axis (see §6.2). The plots in shades correspond to various $H$, and the dark lines to the minimum skyline for each cost factor and $K$ across all values of $H$. The cross-over from SGD to Local SGD happens only for high costs ($m$).

but performance deteriorates for large $K$. In general, using a larger $K$ requires a learning rate increase by factor *smaller* than $K$ and, consequently, training for *more* than $1/K$ times the number of iterations. AdaScale (Johnson et al., 2020) uses estimates of gradient variance and magnitude to compute a local scaling factor for each update step, which is used to modulate the learning rate, as well as to *stretch* the number of training iterations.

While data-parallel training can significantly speed up model training, it necessitates synchronization among workers at each iteration. As the number of workers increases, the communication time required to synchronize begins to dominate the computation time (see Ortiz et al., 2021, Figure 1). This is exacerbated by compute infrastructure with low bandwidth and/or high-latency connections between workers. A straightforward remedy for communication overhead is gradient accumulation, where each worker averages gradients from $H > 1$ minibatches locally before synchronization. However, this increases the effective batch size by a factor of $K \times H$, which quickly enters a regime of diminishing returns. It has been shown to deteriorate performance, even for moderate values of $H$, when using linear learning rate scaling (Lin et al., 2020).

An alternative approach to reduce communication overhead is Local SGD (Stich, 2019; Yun et al., 2022), which performs $H > 1$ gradient steps locally on each worker, after which the model weights are averaged and synchronized across $K$ workers. While theoretical work has shown that Local SGD converges at the same rate as synchronous data-parallel SGD, empirical studies (Lin et al., 2020; Ortiz et al., 2021) have observed deteriorating model performance when using Local SGD.

In this paper, we study this apparent discrepancy between the theory and empirical findings on Local SGD. We, specifically, answer the following three questions:

1. *Why does Local SGD's performance lag that of SGD?* Empirical works that study the performance of Local SGD use learning rate scaling methods borrowed from SGD literature (see §2). We surface that this is problematic for Local SGD. As in the synchronous case, increasing the number of workers reduces the variance of the virtual gradient estimate used by Local SGD, which warrants a learning rate increase. As we show, Local SGD's gradient estimate is *biased*, and the bias depends on the learning rate itself. Thus, a naive application of known learning rate scaling techniques may adversely affect the quality of Local SGD's gradient estimate and, thus, its convergence behavior; this explains the results by Lin et al. (2020); Ortiz et al. (2021) who use linear learning rate scaling and show that Local SGD performs worse than standard SGD (see §4).

2. *Can the performance gap between Local SGD and SGD be bridged?* We examine the optimal learning rates for Local SGD and SGD, and devise automatic learning rate scaling methods that scale the learning rate based on gradient statistics. In this process, we recover the well-known technique for SGD called AdaScale (Johnson et al., 2020) that was previously proposed as a heuristic (see §5). We show that using our proposed scaling technique, Local SGD reliably maintains the target accuracy across a wide range of values for $K$ and $H$ as shown in Figure 1 in blue. This is in contrast to previous works that observe deteriorating performance when using Local SGD with linear learning rate scaling as shown in the green curve in Figure 1 (see §6.1).

3. *When is Local SGD empirically preferable to SGD?* When using automatic learning rate scaling methods, we show that Local SGD converges faster than large batch SGD realized through gradient accumulation for large $H$. However, this does not trivially translate into wall-clock speedups, as these automatic learning rate scaling methods also modulate the number of training iterations. In Figure 2, we show that Local SGD improves wall-clock time convergence compared to synchronous data-parallel SGD, when comparing under optimal learning rate scaling, only in extreme scenarios of communication being substantially more time-consuming than computation. This finding casts fundamental aspersions on the practicality of Local SGD especially when scaling only the learning rate, one we find missing in prior works (see §6.2).

## 2 Related Work

The literature on distributed optimization is vast, so we focus on the most closely related works. For instance, we do not discuss fully asynchronous methods like HogWild (Recht et al., 2011) or approaches to compress gradients for communication (*e.g.*, Alistarh et al., 2017; Basu et al., 2019). Cao et al. (2023) present a survey of such methods.

**Learning rate scaling** To our knowledge, the first practical recommendation for learning rate modulation in distributed optimization was proposed by Krizhevsky (2014), who introduced a *linear scaling rule* in terms of the number of workers. They found it to work well for small numbers of workers but observed performance drops for larger numbers. Goyal et al. (2018) showed that linear scaling could be made to work at larger scales with the use of a warmup phase that gradually increases the learning rate towards the target value. They found the duration of the warm-up to be critical to the performance. As an alternative to linear scaling, Krizhevsky (2014) as well as Hoffer et al. (2017) experimented with a less aggressive square-root heuristic, which ultimately did not prove successful. Going beyond simple scaling factors, AdaScale (Johnson et al., 2020) adaptively scales the learning rate based on estimates of the gradient variance and magnitude and has been shown to retain performance for a much higher number of workers. We will discuss this method in detail in §5. Methods for improved generalization, like cyclical learning rate (Smith, 2017), can be seen orthogonal to our work, as we investigate optimal scaling factors for the learning rates.

**Large-batch optimizers** Optimizers like LARS (You et al., 2017) and LAMB (You et al., 2020) have been proposed with the explicit goal of handling large batch sizes, but their utility has been questioned. Nado et al. (2021) find that, with suitable hyperparameters, both SGD and Adam match the performance of LARS and LAMB. Based on the assumption that small batch training generalizes better, Adasum (Maleki et al., 2021) tries to mimic small batch training while using larger batch sizes using a more complex operation to fuse the gradients instead of summation. Refuting the claims that the stochastic nature of SGD is important

for its performance, Geiping et al. (2022) show that with explicit regularization, even full-batch gradient descent can attain the performance of small batch SGD.

**Local SGD**  The method now known as LocalSGD has been studied both empirically (Povey et al., 2014; Zhang et al., 2016; Lin et al., 2020; Gu et al., 2023), and theoretically (Dekel et al., 2012; Zhou & Cong, 2018; Stich, 2019; Haddadpour et al., 2019; Khaled et al., 2020; Woodworth et al., 2020; Deng et al., 2022; Yun et al., 2022). Deep learning models trained using Local SGD have been shown to achieve lower performance than using fully synchronous training. Lin et al. (2020) remedy this by post-Local SGD, where they switch to Local SGD after training the network with synchronous SGD for a certain number of iterations. Ortiz et al. (2021) show that the point of the switch is a sensitive hyperparameter. They change the global averaging step to a moving average as in Wang et al. (2020) and find it partially alleviates that sensitivity. Note that both Lin et al. (2020) and Ortiz et al. (2021) attempt to achieve perfect linear scaling (*i.e.*, reducing the number of iterations by a factor of $K$) and apply the linear learning rate scaling rule of Goyal et al. (2018) to Local SGD. A closely related work by Wang & Joshi (2019) shows that the number of local steps $H$ needs to be decreased as model reaches closer to convergence. They consider a fixed known learning rate, and optimize $H$, whereas we do the opposite. Thus, the two approaches are flip-sides of the same coin. Recently, Gu et al. (2023) find that Local SGD performs like SGD when using a small learning rate and training *long enough*. Hence, this scenario while useful for analysis, is of limited utility, as in practice we are interested in getting a high enough performance in the shortest time possible.

**Federated learning**  A plethora of Local SGD variants have been used and studied in federated learning (Wang et al., 2021), where workers access fixed subsets of data, which are not necessarily iid. Under these conditions of data heterogeneity, Murata & Suzuki (2021) remark that the gradient estimate of Local SGD is biased but do not quantify it. Works in federated learning that examine learning rate scaling have focussed on heuristics like linear and square-root scaling presented above (*e.g.*, Charles et al., 2021). Data heterogeneity poses additional challenges compared to our setting and is not considered here.

## 3 Preliminaries

In this section, we introduce some preliminary material and define notation.

### 3.1 Problem Setup

Training a neural network requires solving an empirical risk minimization problem with an objective of the following form:

$$f(\mathbf{w}) = \frac{1}{N} \sum_{i=1}^{N} f(\mathbf{w}; \mathbf{x}_i), \tag{1}$$

where $\mathbf{w}$ are the model weights and $f(\cdot; \mathbf{x}_i)$ denotes the loss of the $i$-th training example. Throughout this paper, we assume $f$ to be $L$-smooth, *i.e.*, its gradient is Lipschitz: $\|\nabla f(\mathbf{w}') - \nabla f(\mathbf{w})\| \leq L\|\mathbf{w}' - \mathbf{w}\|$ for all $\mathbf{w}, \mathbf{w}'$ in the domain of $f$. This implies the well-known local quadratic bound $f(\mathbf{w}') \leq f(\mathbf{w}) + \nabla f(\mathbf{w})^T(\mathbf{w}' - \mathbf{w}) + \frac{L}{2}\|\mathbf{w}' - \mathbf{w}\|^2$, which we will use in our analysis.

Gradient descent iteratively minimizes the objective with updates of the form $\mathbf{w}_{t+1} = \mathbf{w}_t - \gamma_t \nabla f(\mathbf{w}_t)$, where $\gamma_t$ is the learning rate at iteration $t$. For large datasets, it is inefficient to compute the gradient over all data points in each iteration. Instead, we resort to Stochastic Gradient Descent (SGD), where we approximate the gradient using a stochastic gradient

$$\mathbf{g}_t = \frac{1}{B} \sum_{i \in \mathcal{B}} \nabla f(\mathbf{w}_t; \mathbf{x}_i), \tag{2}$$

computed on a randomly-sampled minibatch $\mathcal{B} \subset \{1, 2 \cdots N\}$ of size $|\mathcal{B}| = B \ll N$.

In synchronous data-parallel SGD, each worker computes a stochastic gradient $\mathbf{g}_t^k$ using an independent minibatch. These are then averaged, $\tilde{\mathbf{g}}_t = \frac{1}{K}\sum_{k=1}^{K}\mathbf{g}_t^k$, before an optimization step is applied. Each $\mathbf{g}_t^k$ is an unbiased estimate of the gradient, $\mathbb{E}_t[\mathbf{g}_t^k] = \nabla f_t := \nabla f(\mathbf{w}_t)$ with a variance that we denote as $\sigma_t^2 :=$ $\mathbf{Var}_t[\mathbf{g}_t^k]$. Consequently, the averaged gradient $\tilde{\mathbf{g}}_t$ unbiasedly estimates $\nabla f_t$ with a variance of $\sigma_t^2/K$. Here and throughout this paper, $\mathbb{E}_t$ denotes the conditional expectation given $\mathbf{w}_t^k$, $\forall k \in \{1, 2 \cdots K\}$.

## 3.2 Local SGD

Synchronous data-parallel SGD is equivalent to each worker taking a local gradient step, followed by an averaging of the model weights. This requires a synchronization of the weights after every iteration, which may cause a large communication overhead, depending on the number of workers and the compute infrastructure.

In Local SGD, each worker takes $H > 1$ local gradient steps before the weights are averaged:

$$\mathbf{w}_{t+1}^k = \begin{cases} \frac{1}{K}\sum_{k=1}^{K}(\mathbf{w}_t^k - \gamma_t\mathbf{g}_t^k), & \text{if } H \mid t+1, \\ \mathbf{w}_t^k - \gamma_t\mathbf{g}_t^k, & \text{otherwise.} \end{cases} \tag{3}$$

This reduces the communication frequency without affecting the rate of convergence (Stich, 2019). In practice, each of those $H$ iterations can be replaced by a more complex update step, such as using momentum or even adaptive gradient methods like Adam (Singh et al., 2021; Wang et al., 2020).

## 4 Local SGD's Learning Rate Conundrum

The analysis of Local SGD is based on the virtual sequences

$$\bar{\mathbf{w}}_t := \frac{1}{K}\sum_{k=1}^{K}\mathbf{w}_t^k, \quad \bar{\mathbf{g}}_t := \frac{1}{K}\sum_{k=1}^{K}\mathbf{g}_t^k, \tag{4}$$

which are the averages of the per-worker iterates and gradients at each iteration. These sequences are tools for the mathematical analysis and are *not* computed by Local SGD at every iteration. They evolve as $\bar{\mathbf{w}}_{t+1} = \bar{\mathbf{w}}_t - \gamma_t\bar{\mathbf{g}}_t$, resembling an SGD trajectory with an "implicit" gradient estimate $\bar{\mathbf{g}}_t$.

Theoretical works on Local SGD show that this sequence behaves almost like synchronous data-parallel SGD under certain restrictions on the learning rate. This indicates that the implicit gradient estimate $\bar{\mathbf{g}}_t$ approximates the gradient $\nabla f(\bar{\mathbf{w}}_t)$ *just as well* as a hypothetical standard stochastic gradient $\tilde{\mathbf{g}}_t$ with $\mathbb{E}_t[\tilde{\mathbf{g}}_t] = \nabla f(\bar{\mathbf{w}}_t)$ and a variance of

$$\mathbb{E}_t[\|\tilde{\mathbf{g}}_t - \nabla f(\bar{\mathbf{w}}_t)\|^2] = \frac{\sigma_t^2}{K}. \tag{5}$$

The meaning of "just as well" is relatively opaque in existing works on Local SGD. Here, we make it explicit. As we will see, the quality of $\bar{\mathbf{g}}_t$ is influenced by the preceding steps taken since the most recent synchronization point. We, therefore, consider a single constant step size $\gamma$ across $H$ consecutive steps, starting from a synchronization point, and assume a local bound on the gradient variance. This is formalized as follows.

**Assumption 4.1.** Assume we run $H$ consecutive steps of Local SGD using a constant step size $\gamma$, starting from a synchronization point $t : H \mid t$. Assume the gradient variance across these $H$ steps is bounded, *i.e.*, for all $k \in [K]$ and $t' \in [t, t+H)$, we have $\mathbf{Var}_{t'}[\mathbf{g}_{t'}^k] \leq \bar{\sigma}_t^2$ for some $\bar{\sigma}_t$ .

With that, we can derive a bound on the mean-squared error (MSE) of Local SGD's implicit gradient estimate $\bar{\mathbf{g}}$. The proof is given in Appendix C.1.

**Proposition 4.2.** *Under Assumption [4.1], the MSE of Local SGD's implicit gradient estimate (Eq. [4]) satisfies*

$$\mathbb{E}_t \left[ \|\bar{\mathbf{g}}_{t'} - \nabla f(\bar{\mathbf{w}}_{t'})\|^2 \right] \leq \underbrace{\frac{\bar{\sigma}_t^2}{K}}_{\textit{Variance}} + \underbrace{(H-1)L^2\gamma^2\bar{\sigma}_t^2}_{\textit{Bias}} \tag{6}$$

*for all $t' \in [t, t+H)$.*

The bias originates from the fact that the per-worker gradients, $\mathbf{g}_t^k$ in Equation (4), are computed at slightly different locations in the parameter space due to differences in the local trajectories. The extent of this diffusion depends on the number of local steps $H$ and, crucially, the learning rate $\gamma$ used in preceding steps.

The published convergence theory for Local SGD assumes restrictions to the learning rate $\gamma$ which control the bias term such that the error is dominated by the variance term and its $1/K$ behavior. Therefore, those results have to be understood in the following way: For a given value of $H$, there is a small enough learning rate $\gamma$ at which Local SGD converges at the same speed as synchronous data-parallel SGD. However, this learning rate will *not* be optimal, neither for synchronous data-parallel SGD nor for Local SGD.

Equation (6) shows that Local SGD faces a fundamentally different trade-off when setting the learning rate compared to synchronous data-parallel SGD. Increasing $K$ decreases the variance, which warrants a learning rate increase. However, a larger learning rate increases the bias term.

Empirical studies (Lin et al., 2020; Ortiz et al., 2021) observing the poor practical performance of Local SGD for large values of $H$ could in part be explained by this. Since they adopt the linear scaling rule for Local SGD, for large values of $K$ and $H$, the bias term will dominate the total error in Equation (6). The same argument explains the observations of Wang & Joshi (2019); when the learning rate is fixed, the only way of reducing the gradient error (and therefore facilitating convergence) is to reduce the communication interval $H$, leading to their heuristic. In the next section, we propose a learning rate scaling rule for Local SGD that directly tackles the trade-off surfaced in Equation (6).

## 5 Adaptive Learning Rate Scaling for Synchronous and Local SGD

To gauge the real potential of Local SGD, we need to be able to scale its learning rate appropriately. We have seen in the previous section that Local SGD faces a fundamentally different trade-off with respect to the learning rate, compared to synchronous SGD. In this section, we first derive an optimal learning rate scaling technique for SGD. We then carry-over the ideas to derive an optimal learning rate scaling technique for Local SGD.

### 5.1 Optimal learning rate scaling for SGD – AdaScale

From the assumption of Lipschitz gradients in §3.1, $f$ satisfies

$$\mathbb{E}[f(\mathbf{w}_{t+1})] \leq f(\mathbf{w}_t) - \gamma_t \nabla f_t^T \cdot \mathbb{E}[\mathbf{g}_t] + \frac{L \cdot \gamma_t^2}{2}\mathbb{E}[\|\mathbf{g}_t\|^2] \tag{7}$$

for two consecutive iterates $\mathbf{w}_t$, $\mathbf{w}_{t+1}$ with an SGD step. The optimal learning rate that maximizes the expected decrease over one step is given by

$$\gamma_t = \frac{1}{L} \cdot \frac{\|\nabla f_t\|^2}{\|\nabla f_t\|^2 + \sigma_t^2}. \tag{8}$$

The optimal learning rate requires $L$ which is difficult to estimate in practice, and thus not used in practice. Consequently, when the variance of our gradient estimate is reduced by a factor of $1/K$ due to an increase in number of workers to $K$, the optimal learning rate changes by the following factor:

$$r_t = \frac{\|\nabla f_t\|^2 + \sigma_t^2}{\|\nabla f_t\|^2 + \frac{\sigma_t^2}{K}} \in [1, K). \tag{9}$$

This is the *gain ratio* proposed by Johnson et al. (2020). Thus, we reinterpret the AdaScale gain ratio as the ratio of optimal learning rates for the case of $K$ workers to 1 worker. We present a detailed derivation in Appendix B.

For $\|\nabla f_t\|^2 \ll \sigma_t^2$, we recover the linear scaling rule of Goyal et al. (2018) with a gain ratio of $r_t \approx K$. In practice, however, the gain ratio will be smaller than $K$. When $\|\nabla f_t\|^2 \gg \sigma_t^2$, the gradient estimate is already very accurate, and the gain ratio is close to 1, implying that using a higher number of workers has no benefit. This is related to the concept of a critical batch size (McCandlish et al., 2018), which is the point after which a further increase in batch size has diminishing returns in terms of convergence speed.

In addition to using $r_t$ as a scaling factor for the learning rate, Johnson et al. (2020) propose the concept of *scale-invariant iterations*. When using the standard linear scaling rule (Goyal et al., 2018), the iteration counter is incremented by a factor of $K$ for a forward-backward pass. However, AdaScale interprets the gain ratio $r_t$ to be the effective number of workers used at iteration $t$. Incorporating this into the training process, they maintain an accumulator $s_t = \sum_{t'=0}^{t-1} r_{t'}$, which replaces the standard iteration counter. Since $r_t < K$, the use of scale-invariant iterations increases the total number of passes over the training set (true epochs) and "stretches" the learning rate schedule accordingly; see Lines 1 and 7 of Algorithm 2 in the Appendix. Thus, scaling to a larger number of workers may necessitate more true epochs but will typically still result in a substantial wall-clock time speedup, as shown by Johnson et al. (2020).

AdaScale estimates $\|\nabla f_t\|$ and $\sigma_t^2$ from the $K$ iid per-worker gradients. We adopt their estimation procedure for our scaling method for Local SGD.

### 5.2 Optimal learning rate scaling for Local SGD– LocalAdaScale

We now derive a learning rate scaling method similar to AdaScale for Local SGD, named LocalAdaScale using the same principle underlying our derivation of AdaScale. We first derive an optimal step size for Local SGD, depending on the number of workers $K$ and local steps $H$. A gain ratio is obtained by dividing by the optimal step size for the base case ($K = H = 1$) and will be used exactly as in AdaScale, both as a scaling factor for the learning rate, as well as the basis for a scale-invariant iteration counter.

**Optimal step size for Local SGD** Finding the optimal step size for Local SGD is more complicated than for synchronous data-parallel SGD since the step size used in previous steps influences the quality of the (implicit) gradient estimate used in the current step. To simplify this, we adopt Assumption 4.1 from §4 and consider $H$ consecutive steps using a constant step size. As we show in Appendix C.2, these steps lead to an expected decrease in function value, which is bounded as follows:

$$\mathbb{E}_t[f(\bar{\mathbf{w}}_{t+H})] \le f(\bar{\mathbf{w}}_t) - H\left(\frac{\gamma}{2}\bar{G}_t + \frac{\gamma}{2}\bar{A}_t - \frac{\gamma^2 L}{2}\bar{A}_t - \frac{\gamma^2 L}{2}\frac{\bar{\sigma}_t^2}{K} - \frac{\gamma^3 L^2}{4}(H-1)\bar{\sigma}_t^2\right), \tag{10}$$

where

$$\bar{G}_t = \frac{1}{H}\sum_{t'=t}^{t+H-1}\mathbb{E}_t[\|\nabla f(\bar{\mathbf{w}}_{t'})\|^2], \quad \bar{A}_t = \frac{1}{K}\sum_k \frac{1}{H}\sum_{t'=t}^{t+H-1}\mathbb{E}_t[\|\nabla f(\mathbf{w}_{t'}^k)\|^2]. \tag{11}$$

This expected function decrease bound features the cumbersome terms $\bar{G}_t$ and $\bar{A}_t$, which are expected squared gradient magnitudes along the trajectories of the virtual averaged iterates, and the per-worker iterates, respectively. These quantities are, in principle, dependent on $\eta$, since the choice of step size influences the gradient magnitude along that trajectory. We ignore this secondary effect by assuming $\bar{A}_t \approx \|\nabla f(\bar{\mathbf{w}}_t)\|^2 \approx \bar{G}_t$, independent of $\eta$. This is fulfilled if the gradient magnitude stays approximately constant over the $H$ subsequent steps. We verify this experimentally in Figure 7 in the Appendix. Using this assumption gives us an *approximate* upper bound:

$$\mathbb{E}_t[f(\bar{\mathbf{w}}_{t+H})] \overset{\approx}{\le} f(\bar{\mathbf{w}}_t) - H\underbrace{\left(\gamma\bar{G}_t - \frac{\gamma^2 L}{2}\bar{G}_t - \frac{\gamma^2 L}{2}\frac{\bar{\sigma}_t^2}{K} - \frac{\gamma^3 L^2}{4}(H-1)\bar{\sigma}_t^2\right)}_{\otimes}. \tag{12}$$

In the expected decrease bound in Equation (10), the bias discussed in §4 manifests as the $\gamma^3$ term. To illustrate that, Appendix D provides a similar $H$-step bound for synchronous data-parallel SGD, where this term is absent.

We derive an approximately optimal step size for Local SGD by maximizing the term $\bigotimes$ in Equation (12) leading to

$$\gamma_t = \frac{1}{L} \cdot \frac{2\bar{G}_t}{\bar{G}_t + \frac{\bar{\sigma}_t^2}{K} + \sqrt{\left(\bar{G}_t + \frac{\bar{\sigma}_t^2}{K}\right)^2 + 3(H-1)\bar{G}_t\bar{\sigma}_t^2}} \tag{13}$$

See Appendix C.3 for a derivation.

**Gain ratio**  Analogous to our derivation of AdaScale, we can now define a gain ratio by dividing the optimal step size in Equation (13) by the base case ($H = K = 1$):

$$\rho_t = \frac{2\left(\bar{G}_t + \bar{\sigma}_t^2\right)}{\bar{G}_t + \frac{\bar{\sigma}_t^2}{K} + \sqrt{\left(\bar{G}_t + \frac{\bar{\sigma}_t^2}{K}\right)^2 + 3(H-1)\bar{G}_t\bar{\sigma}_t^2}}. \tag{14}$$

When $H = 1$, this gain ratio recovers AdaScale. To illustrate the difference in behaviors between AdaScale and LocalAdaScale, we plot the gain ratio $\rho$ in Figure 3 for increasing values of $\bar{\sigma}^2/\bar{G}$. The larger the $H$, the smaller the gain ratio at any given value for $\bar{\sigma}^2/\bar{G}$. This is explained by the fact that the gradient estimates get worse with $H$ as seen in Proposition 4.2. Looking at the inset plot: for a deterministic case, *i.e.*, when there is no variance ($\bar{\sigma}^2 \to 0$) and thus no advantage to using multiple workers, we see that $\rho = 1$ for AdaScale and LocalAdaScale. For a large variance $\bar{\sigma}^2 \gg \bar{G}$, the gain ratio approaches $K$ for all values of $H$, recovering the linear scaling rule. However, for larger $H$, it takes substantially higher values of $\bar{\sigma}^2/\bar{G}$ for that regime to be reached.

---

**Algorithm 1** Automatic learning rate scaling for Local SGD– LocalAdaScale.

---

**Input:** Initialization $\mathbf{w}_0$, step-size $\gamma_t$, #workers $K$, #local steps $H$, scale-inv budget $\mathcal{S}$, $t = 0$, $s = 0$, grad_cache=[], $\rho \leftarrow 1$.

1: **while** $s \leq \mathcal{S}$ **do**  ▷ Scale inv budget not exhausted.
2:     **for** $k \in [K]$ **do**  ▷ On each worker
3:         Compute $\mathbf{g}_t^k$ using a batch of data. ▷ Gradient at $t$.
4:         **if** $H \mid t$ **then**  ▷ One step after model sync.
5:             grad_cache$[k] = \mathbf{g}_t^k$.  ▷ Save gradient.
6:         **if** $H \mid (t+1)$ **then**  ▷ Average every $H$ steps
7:             $\mathbf{w}_t^k \leftarrow \frac{1}{K}\sum_{j=1}^K \mathbf{w}_t^j$.
8:         $\bar{G}_t, \bar{\sigma}_t^2 \leftarrow$ grad_stats(grad_cache)  ▷ Eq.15
9:         Compute $\rho$ as Equation (14).
10:         $\mathbf{w}_{t+1}^k \leftarrow \mathbf{w}_t^k - \rho\gamma_{\lceil s\rceil}\mathbf{g}_t^k$.  ▷ Local update.
11:     $s \leftarrow s + \rho$
12:     $t \leftarrow t + 1$
13: **return** the last iterate $\mathbf{w}_t$.

---

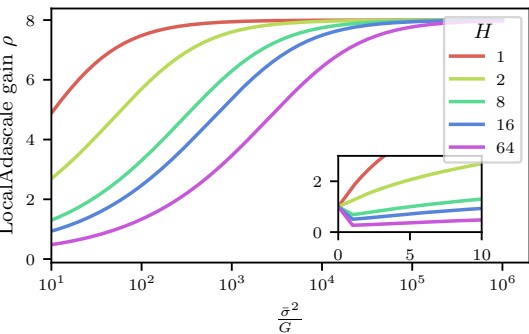

Figure 3: LocalAdaScale gain ratio from Equation (14) with $K = 8$. We plot $\bar{\sigma}^2/\bar{G}$ on the x-axis and the gain ratio on the y-axis. In the inset, we show the plot for very small values of the ratio $\bar{\sigma}^2/\bar{G}$. We see that synchronous SGD (AdaScale with $H = 1$) and Local SGD (LocalAdaScale $H > 1$) have different scaling behaviour. AdaScale reaches its maximum gain factor for much lower values of $\bar{\sigma}^2/\bar{G}$ than LocalAdaScale. In practice, this translates to Local SGD needing a lower learning rate, and more iterations than SGD. When this is violated, Local SGD exhibits poorer convergence behaviour, further evidencing the arguments in §4

**Implementing LocalAdaScale** To implement LocalAdaScale, we need to estimate $\bar{G}_t$ and $\bar{\sigma}_t^2$. We take the approach of estimating these quantities only at synchronization points, and then using the resulting gain throughout the following $H$ local steps. This is in line with our assumption that gradient magnitude and variance are approximately constant over $H$ steps. At a synchronization point $t$, we have access to $K$ per-worker gradients computed at the *same* location. This allows us to estimate gradient variance and mean exactly as in AdaScale:

$$\bar{\sigma}_t^2 \approx \frac{1}{K-1} \sum_{k=1}^{K} \|\mathbf{g}_t^k\|^2 - \frac{K}{K-1} \|\bar{\mathbf{g}}_t\|^2, \quad \bar{G}_t \approx \|\bar{\mathbf{g}}_t\|^2 - \frac{1}{K} \bar{\sigma}_t^2. \tag{15}$$

Following AdaScale, these estimates are smoothed using an exponential moving average for additional stability. The algorithms LocalAdaScale and AdaScale are summarized in Algorithm 1, as LocalAdaScale is equivalent to AdaScale when $H = 1$.

When using Local SGD, computing the estimates in Equation (15) requires the synchronization of gradients in addition to the weights, doubling the amount of data communicated. We partially alleviate this issue as follows. We cache the gradients computed right after synchronization and delay their communication (and thereby the computation of the gain ratio) until the *next* synchronization step.[1] This way, we can synchronize weights and gradients simultaneously and only incur the *latency* overhead once. Of course, this does not alleviate communication time incurred due to *bandwidth* limitations. Note, however, that we devised LocalAdaScale primarily as a tool for the analysis of Local SGD and did not further optimize our implementation. In future work, one could devise approximate versions of LocalAdaScale that avoid additional communication.

## 6 Experiments

Our experiments compare Local SGD to the gradient accumulation baseline under two scaling approaches, amounting to a total of four different methods. We compare these methods under identical numbers of workers ($K$) and communication/synchronization intervals $H$:

1. Gradient accumulation of $H$ steps with linear scaling (ACC-LIN).
2. Gradient accumulation of $H$ steps with AdaScale (ACC-ADA).
3. Local SGD with $H$ local steps with linear scaling (LOCAL-LIN).
4. Local SGD with $H$ local steps adaptive scaling (LOCAL-ADA), the method derived in §5.

These four methods process the same number of samples between two consecutive synchronizations. The linear scaling variants perform exactly $n$ true epochs, where $n$ is set based on prior work for each model/dataset combination (see Appendix H.1). The adaptive scaling variants are allocated a budget of $n$ *scale-invariant* epochs. Recall from §5.1 that this results in a variable number of true epochs.

For linear scaling, we follow prior work and perform a linear warmup from $\gamma^{\text{base}}$ to the final value of $K\gamma^{\text{base}}$ for 5% of the total iteration budget. For AdaScale, we use the implementation from FairScale (FairScale authors, 2021) which supports gradient accumulation. We compare these methods at different values for the number of workers $K$, as well as the communication interval $H$. The latter is used as the number of local steps or accumulation steps, respectively.

We train a ResNet-18 (He et al., 2016) on CIFAR-10 (Krizhevsky, 2009), a WideResnet-28-2 (Zagoruyko & Komodakis, 2016) on ImageNet-32 (Chrabaszcz et al., 2017), and a ResNet-50 on ImageNet (Deng et al., 2009; Russakovsky et al., 2015). We use a base learning rate schedules from previously published works, which are well-tuned to the respective architecture and dataset, which we take to be the optimal learning rates for the base case *i.e.*, $K = 1, H = 1$. Note that we are comparing learning rate *scaling* techniques and not specific learning rate schedules. Additional details of our experimental setup can be found in Appendix H.1.

---

[1]In Appendix E, we perform an ablation study showing that this delayed communication of gradients does not alter the behavior of the method significantly.

### 6.1 Local SGD with automatic learning rate scaling LocalAdaScale maintains Target Accuracy

In Figure 4 we show the test accuracy reached by the four methods. The target performance for each experiment is the performance we get when training on one worker ($K = 1$) with the standard hyperparameter settings (in Appendix H). For CIFAR-10 it is a top-1 accuracy of 93%, for ImageNet32 it is a top-5 accuracy of 69%, and for ImageNet it is a top-5 accuracy of 93%. See Appendix H.1 for attributions.

We see that ACC-LIN breaks down very quickly for all datasets. Performance starts dropping when the total scaling ($K \times H$) exceeds 32, in line with the findings by Goyal et al. (2018). This is overcome by using AdaScale (ACC-ADA), which maintains the target accuracy even for large $H$ across all our experiments. In the Local SGD family, LOCAL-LIN performs considerably better but we still see notable drops in test accuracy around $H \geq 8$. Finally, adaptive learning rate scaling for Local SGD maintains the target performance across all $K$ and $H$ considered. Thus, we have demonstrated that appropriate learning rate scaling can fix the performance gap of Local SGD observed in previous works using LOCAL-LIN (Lin et al., 2020; Ortiz et al., 2021).

### 6.2 Should I Use Local SGD?

We have established with the results in Figure 4 that both Local SGD and gradient accumulation can maintain high accuracy in the presence of appropriate learning rate scaling. As described in §5, ACC-ADA and LOCAL-ADA not only modulate the learning rate but also increase the number of iterations by using scale-invariant epochs. Therefore the question of whether the reduced communication time makes up for the increased number of iterations arises.

Our experiments help answer this question. In Figure 5, we compare the number of iterations required by each method. We see that increasing $H$, both adaptive scaling methods increase the number of iterations quite drastically. For $H < 8$, ACC-ADA converges in slightly fewer iterations than LOCAL-ADA, whereas LOCAL-ADA is more iteration efficient for more infrequent communication (larger $H$). See also the tabulated results in Appendix H for details.

But which $H$ should we choose in practice? Answering this question requires us to make assumptions about the relative cost of communication and computation. We let $m$ denote the relative communication overhead, *i.e.*, the ratio of time taken for one communication round to the time taken for one minibatch gradient computation without any synchronization. This is determined by the hardware and, therefore, beyond our control on the algorithmic side. Given the total number of epochs $n^{(K,H)}$ observed in our experiments for LOCAL-ADA and ACC-ADA (Figure 5), we compute a pseudo-wall-clock training time as follows:

$$T^{(K,H)}_{\text{method}} = \underbrace{\frac{n^{(K,H)}_{\text{method}}}{K}}_{\text{\# Iterations}} \times \left( \underbrace{1}_{\text{Computation}} + \overbrace{\frac{m}{H}}^{\text{Communication}} \right). \tag{16}$$

In Equation (16), we do not show dependence on per-device batch size $B$ and dataset size explicitly in the term for number of iterations as they are constant scalars across methods. We plot this wall-clock time in Figure 6 for CIFAR-10 and ImageNet32 at $K = 24$ and $H \in \{1, 2, 8\}$; additional results may be found in Appendix H.3. For CIFAR-10, we see that SGD is faster than Local SGD for all $H$ when $m \leq 3$, and Local SGD otherwise. However, a different picture emerges for ImageNet-32; small values of $H$ (say $2, 8$) for both gradient accumulation and Local SGD result in faster convergence than SGD. This is possibly because the overall batch size ($B \times 24$) is smaller than the critical batch size and thus gradient accumulation results in faster convergence. Local SGD is faster than SGD (for some $H$) when the dotted lines are below the solid ones in Figure 6, and we see that Local SGD is faster than SGD (with gradient accumulation) only when $m \geq 4$. Additional plots may be found in Figure 11 in the Appendix. Such large values of $m$ are atypical in most training set-ups. Ortiz et al. (2021) show that even when using $K = 256$, $m$ stays around 3. A substantially larger communication overhead is plausible for some setups, *e.g.*, crowd-sourced

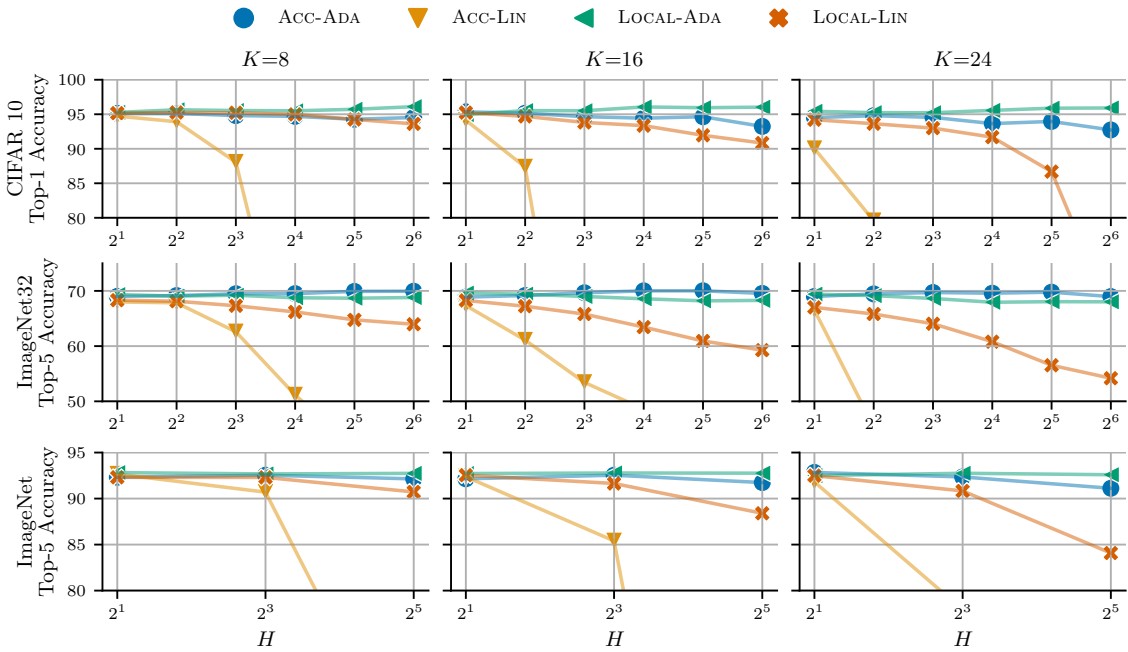

Figure 4: Test accuracies achieved for different numbers of workers ($K$) and communication intervals ($H$). Linear scaling with gradient accumulation deteriorates quickly. Linear scaling with Local SGD is more robust but suffers for large values of $H$. Using adaptive scaling, both gradient accumulation and Local SGD maintain the target performance across a large range of values for $K$ and $H$.

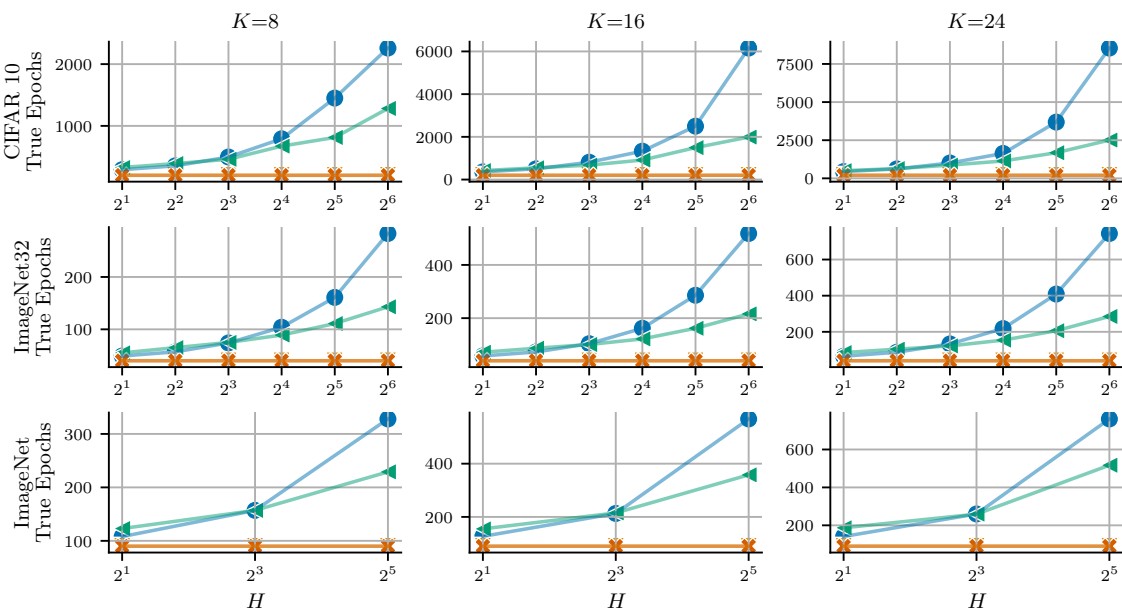

Figure 5: Number of epochs used by each method. While linear scaling operates under a fixed budget, the adaptive scaling methods *stretch* the learning rate schedule, which increases the total number of epochs. For values of $H \leq 8$, ACC-ADA uses slightly fewer epochs than LOCAL-ADA. For large values of $H$, LOCAL-ADA is drastically more iteration-efficient.

or volunteer computing (Ryabinin & Gusev, 2020). However, such scenarios come with a host of other issues like stragglers, fault tolerance that have propelled their own lines of research (Learning@home team, 2020; Borzunov et al., 2022a;b; Ryabinin et al., 2021; Blanchard et al., 2017). Therefore, our findings cast doubts on the practical utility of Local SGD in commonplace distributed training environments.

## 7 Conclusion

**Summary of Findings**  The previously published literature on Local SGD suffers from a certain discrepancy between theory and practice. Previous theoretical results have been interpreted as: "Local SGD behaves just like synchronous data-parallel SGD". In contrast to that, empirical studies have reported performance degradation when using Local SGD compared to synchronous SGD.

We show that this discrepancy may be attributed to the choice of learning rate. Theoretical results have assumed restrictive upper bounds on the learning rate. This recovers the behavior of synchronous SGD at the same learning rate, but that learning rate is clearly non-optimal. Empirical works have used learning rate scaling techniques tailored to synchronous SGD. We show that this is likewise non-optimal, since Local SGD faces a fundamentally different trade-off than synchronous data-parallel SGD where the error of its gradient estimate depends on the learning rate itself.

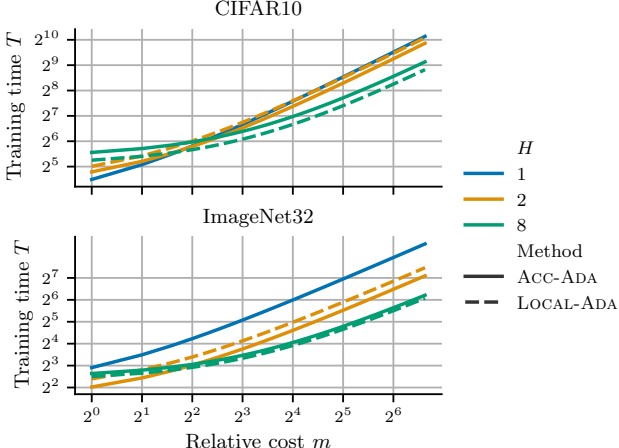

Figure 6:  Which method is faster? We plot pseudo-wall-clock time $T$ on the y-axis for different assumed values of the relative communication overhead $m$, see Eq.(16)). We see that LocalAdaScale (using large values of $H$) converges faster for high communication overheads. For lower $m$, synchronous SGD with few gradient accumulation steps is preferable.

To further study the behavior of Local SGD, we devise an optimal learning rate scaling method, called LocalAdaScale, mirroring the AdaScale method for synchronous SGD. Our experiments demonstrate that LocalAdaScale bridges the gap in performance between Local SGD and synchronous SGD and that Local SGD converges in fewer iterations than gradient accumulation for large communication intervals. However, in wall-clock time, we find that Local SGD is faster than synchronous data-parallel SGD only for very high communication overheads, shedding new light on the practicality of Local SGD.

**Limitations and Future Work**  The optimality of learning rate in this work is based on the training loss and not the test loss. Thus, the claims of optimality have to be seen in the limited context of training performance and optimization, not generalization. Several techniques that target improved generalization (Orvieto et al., 2022) have been investigated, and studying them for Local SGD is beyond the scope of the current work.

Our paper is limited to analyzing the effect of the learning rate while keeping other hyperparameters fixed. Tuning other hyperparameters, such as momentum, may be useful to improve convergence or generalization. We also performed our comparison at a fixed communication interval $H$. A comparison of Local SGD vs. gradient accumulation under varying $H$ (*e.g.*, such as in post-local SGD (Lin et al., 2020)) may be interesting.

The optimal step size for Local SGD in Equation (13) involves multiple approximations. Firstly, like AdaScale, it is based on a Lipschitz bound which may be loose for some objective functions. Secondly, we assume the gradient magnitude along $H$ consecutive Local SGD steps to be approximately constant. Finally, we

estimate gradient magnitude using imperfect empirical estimates, rendering our learning rate scaling only approximately optimal.

We have restricted our analysis to the case of homogeneous data distribution on each worker. An extension of our learning rate scaling technique to the federated learning scenario with heterogeneous data would be interesting future work. Our experiments are limited to computer vision tasks and for models based on convolutional nets. Thus, our findings on the value of Local SGD might be limited to those conditions. Broader generalizations require further experimentation.

As discussed in §5, our implementation of LocalAdaScale synchronizes both the model weights and gradients, which *adds* communication compared to plain Local SGD. We did not attempt to alleviate this, since we devised LocalAdaScale primarily as a tool for the analysis of Local SGD. In future work, one could devise approximate versions of LocalAdaScale that avoid additional communication. A possible avenue would be to approximate gradient magnitude and variance from the *pseudo-gradients* (Reddi et al., 2021) given by the displacement $\overline{\mathbf{w}}_{t+H} - \overline{\mathbf{w}}_t$.

Beyond that, our work can be extended in several ways. While we have compared gradient accumulation and local steps as alternatives, one may realize a desired communication interval $H$ using a combination of both. (For example, 8 local steps, each using 4 gradient accumulations, achieves a communication interval of 32.) Relatedly, one may alter the communication interval for different phases of the optimization process. Both decisions will influence learning rate scaling and may, in turn, be informed by gradient statistics.

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

## Contents

## A Glossary of Symbols

| Symbol | Meaning |
|---|---|
| $K$ | Number of workers |
| $H$ | Number of local steps |
| $\gamma^{\text{base}}$ | Unscaled learning rate for $K = 1$ |
| $B$ | Batch size per worker/Base batch size |
| $(\cdot)_t$ | Quantity $(\cdot)$ at time $t$ |
| $\nabla f_t$ | Gradient of function $f$ at $\mathbf{w}_t$ |
| $\sigma_t^2$ | Variance of gradient estimation at time $t$ |
| $\mathbf{g}_t^k$ | Gradient estimate on worker $k$ at time $t$ |
| $\bar{G}_t, \sigma_t$ | Virtual Gradient magnitude and variance estimate for Local SGD |
| $\mathbf{w}_t^k$ | Model weights on worker $k$ at time $t$ |

Table 1: Glossary of symbols used

## B Derivation of AdaScale

In this section, we derive the AdaScale rule of (Johnson et al., 2020) using ideas from Balles et al. (2017). This view of AdaScale being the learning rate scaling that maximizes the function decrease is novel to the best of our knowledge. Pseudo-code for AdaScale is shown in Algorithm 2.

By definition, $f$ satisfies the following property.

$$f(y) \leq f(x) + \nabla f(x)^T (y - x) + \frac{L}{2}||y - x||^2 \tag{17}$$

for $x, y \in \text{dom}(f)$. Plugging in $x = \mathbf{w}_t$ and $y = \mathbf{w}_{t+1} = \mathbf{w}_t - \gamma_t \mathbf{g}_t$, and computing the expected value of $f(\mathbf{w}_{t+1})$ given $\mathbf{w}_t$ ,where $\mathbf{g}_t$ is the stochastic gradient at $t$, we get

$$\mathbb{E}[f(\mathbf{w}_{t+1})] \leq f(\mathbf{w}_t) - \gamma_t \nabla f_t^T \cdot \mathbb{E}[\mathbf{g}_t] + \frac{L\gamma_t^2}{2} \mathbb{E}[||\mathbf{g}_t||^2] \tag{18}$$

The optimal learning rate is the one that minimizes the right-hand side, thereby maximizing the expected decrease of the function value.

Differentiating the right side of Equation (18) and setting it to zero gives the optimal learning rate as

$$\gamma_t = \frac{\nabla f_t^T \mathbb{E}[\mathbf{g}_t]}{L \cdot \mathbb{E}[||\mathbf{g}_t||^2]} = \frac{1}{L} \frac{\bar{G}_t}{(\sigma_t^2 + \bar{G}_t)} \tag{19}$$

where $\mathbb{E}_t[\mathbf{g}_t] = \nabla f_t$, $||\nabla f_t||^2 := \bar{G}_t$ and $\mathbb{E}_t[||\mathbf{g}_t||^2] = \bar{G}_t + \sigma_t^2$.

When we increase the batch size by a factor of $K$, the variance is reduced by a factor of $K$. Thus, the optimal learning rate becomes

$$\gamma_t^K = \frac{1}{L} \frac{\bar{G}_t}{\left(\frac{\sigma_t^2}{K} + \bar{G}_t\right)}. \tag{20}$$

Thus, the relative change in the learning rate when the batch size is increased $K$ times is

$$r_t = \frac{\sigma_t^2 + \bar{G}_t}{\frac{\sigma_t^2}{K} + \bar{G}_t}, \tag{21}$$

and is termed the gain ratio. As it is evident, it takes a maximum value of $K$ when $\bar{G} \ll \sigma^2$.

---

**Algorithm 2** AdaScale for synchronous gradient descent

---

Input: Initialization $\mathbf{w}$, $t = 0$, learning rate function $\gamma_t$, # workers $K$, scale-invariant iteration budget $\mathcal{S}$, $s = 0$ $t = 0$.

1: **while** $s \leq \mathcal{S}$ **do**            ▷ Scale invariant budget not exhausted
2:     **for** $k \in [K]$ **do**            ▷ On each worker
3:        Compute $\mathbf{g}_t^k$ using a batch of data.        ▷ Compute local gradients.
4:     Compute $r_t \in [1, K]$ as given in Equation (21).
5:     Update $\mathbf{w}_{t+1} \leftarrow \mathbf{w}_t - \gamma_{\lceil s \rceil} \times r_t \tilde{\mathbf{g}}_t$        ▷ $\left( \tilde{\mathbf{g}}_t := \frac{1}{K} \sum_{k=1}^{K} \mathbf{g}_t^k \right)$.
6:     $t \leftarrow t + 1$.            ▷ Iteration counter increment.
7:     $s \leftarrow s + r_t$.            ▷ Budget is computed from the scaling obtained.
8: **return** Last iterate $\mathbf{w}_t$.

---

## C    Details on Local SGD

This section contains details regarding Local SGD which have been omitted from the main text, in particular the derivation of the MSE of its implicit gradient estimate (Appendix C.1) as well as the optimal step size (Appendix C.3).

Local SGD was introduced in §3, in particular Equation (3). In practice, Local SGD is often used with local steps other than a vanilla SGD step, *e.g.*, momentum variants. Algorithm 3 provides pseudocode; Line 5 calls an update function, which may apply any gradient-based optimization step, like SGD or momentum SGD or adaptive gradient methods like Adam.

---

**Algorithm 3** Local SGD

---

Input: Initialization $\mathbf{w}_0$, step-size $\gamma_t$, #workers $K$, #local steps $H$, budget $T$

1: $t \leftarrow 0$
2: **while** $t \leq T$ **do**            ▷ Training budget not exhausted.
3:     **for** $k \in [K]$ **do**            ▷ On each worker
4:        Compute $\mathbf{g}_t^k$ using a batch of data.
5:        $\mathbf{w}_{t+1}^k \leftarrow \text{Update}(\mathbf{w}_t^k, \gamma_t, \mathbf{g}_t^k)$.        ▷ Local update
6:        **if** $t + 1 \mid H$ **then**        ▷ Average every $H$ steps
7:           $\mathbf{w}_{t+1}^k \leftarrow \frac{1}{K} \sum_{k'=1}^{K} \mathbf{w}_{t+1}^{k'}$
8:     $t \leftarrow t + 1$
9: **return** Averaged iterate $\frac{1}{K} \sum_{k=1}^{K} \mathbf{w}_T^k$.

---

### C.1    MSE of Local SGD's Gradient Estimate (Equation 6)

We derive the error in estimating the gradient for Local SGD, stated in Proposition 4.2. We first give the following Lemma, a variant of Lemma 1 in Khaled et al. (2020) using our local variance bound.

**Lemma C.1.** *Let Assumption 4.1 hold and let* $t' \in [t, t + H)$. *Define* $V_{t'} = \frac{1}{K} \sum_{k=1}^{K} \left\| \mathbf{w}_{t'}^k - \bar{\mathbf{w}}_{t'} \right\|^2$. *Then*

$$\mathbb{E}_t[V_{t'}] \leq (H - 1)\gamma^2 \bar{\sigma}_t^2 \tag{22}$$

*Proof.* See Lemma 1 of Khaled et al. (2020).      □

We are now ready to prove Proposition 4.2.

*Proof of Proposition 4.2.* Using the definition of $\bar{\mathbf{g}}_t$ (Equation (4)), for every $t' \in [t, t+H)$, we have

$$\mathbb{E}_{t'}\left[\|\bar{\mathbf{g}}_{t'} - \nabla f(\bar{\mathbf{w}}_{t'})\|^2\right] = \mathbb{E}_{t'}\left[\left\|\frac{1}{K}\sum_{k=1}^{K}(\mathbf{g}_{t'}^k - \nabla f(\bar{\mathbf{w}}_{t'}))\right\|^2\right] \tag{23}$$

$$= \mathbb{E}_{t'}\left[\left\|\frac{1}{K}\sum_{k=1}^{K}\left(\mathbf{g}_{t'}^k - \nabla f(\mathbf{w}_{t'}^k)\right) + \frac{1}{K}\sum_{k=1}^{K}\left(\nabla f(\mathbf{w}_{t'}^k) - \nabla f(\bar{\mathbf{w}}_{t'})\right)\right\|^2\right] \tag{24}$$

$$= \mathbb{E}_{t'}\left[\underbrace{\left\|\frac{1}{K}\sum_{k=1}^{K}\left(\mathbf{g}_{t'}^k - \nabla f(\mathbf{w}_{t'}^k)\right)\right\|^2\right]}_{\text{Term 1}} + \underbrace{\left\|\frac{1}{K}\sum_{k=1}^{K}\left(\nabla f(\mathbf{w}_{t'}^k) - \nabla f(\bar{\mathbf{w}}_{t'})\right)\right\|^2}_{\text{Term 2}} \tag{25}$$

$$\tag{26}$$

For Term 1, the variance bound in Assumption 4.1 gives us

$$\mathbb{E}_{t'}\left[\left\|\frac{1}{K}\sum_{k=1}^{K}\left(\mathbf{g}_{t'}^k - \nabla f(\mathbf{w}_{t'}^k)\right)\right\|^2\right] = \frac{1}{K^2}\sum_{i,j}\underbrace{\mathbb{E}[(\mathbf{g}_{t'}^i - \nabla f(\mathbf{w}_{t'}^i))^T(\mathbf{g}_{t'}^j - \nabla f(\mathbf{w}_{t'}^j))]}_{\leq \bar{\sigma}_t^2 \delta_{ij}} \leq \frac{\bar{\sigma}_t^2}{K}. \tag{27}$$

For Term 2, we use Jensen's inequality and $L$-smoothness to get

$$\left\|\frac{1}{K}\sum_{k=1}^{K}\left(\nabla f(\mathbf{w}_{t'}^k) - \nabla f(\bar{\mathbf{w}}_{t'})\right)\right\|^2 \leq \frac{1}{K}\sum_{k=1}^{K}\left\|\left(\nabla f(\mathbf{w}_{t'}^k) - \nabla f(\bar{\mathbf{w}}_{t'})\right)\right\|^2 \leq L^2 \underbrace{\frac{1}{K}\sum_{k=1}^{K}\left\|\mathbf{w}_{t'}^k - \bar{\mathbf{w}}_{t'}\right\|^2}_{=:V_{t'}}. \tag{28}$$

Plugging that back in and taking the expectation $\mathbb{E}_t$, we find

$$\mathbb{E}_{t'}\left[\|\bar{\mathbf{g}}_{t'} - \nabla f(\bar{\mathbf{w}}_{t'})\|^2\right] \leq \frac{\bar{\sigma}_t^2}{K} + L^2\mathbb{E}_t[V_{t'}] \leq \frac{\bar{\sigma}_t^2}{K} + (H-1)L^2\gamma^2\bar{\sigma}_t^2. \tag{29}$$

$$\square$$

## C.2 Expected Decrease (Equation 10)

We now derive an upper bound on the expected decrease achieved by $H$ consecutive steps of Local SGD. This result was stated in Equation (10). The following Proposition states the bound, where we factorize the learning rate as $\gamma = \frac{\eta}{L}$; this is solely for the readability of the proof.

**Proposition C.2.** *Let $f$ be $L$-smooth. We consider $H$ consecutive steps of Local SGD (Equation (3)) starting from a synchronization point $H \mid t$. We assume a fixed step size $\gamma = \eta/L$ across these $H$ steps. Then*

$$\mathbb{E}_t[f(\bar{\mathbf{w}}_{t+H})] \leq f(\bar{\mathbf{w}}_t) - H\left(\frac{\eta}{2L}\bar{G}_t + \frac{\eta}{2L}\bar{A}_t - \frac{\eta^2}{2L}\bar{A}_t - \frac{\eta^2}{2L}\frac{\bar{\sigma}_t^2}{K} - \frac{\eta^3}{4L}(H-1)\bar{\sigma}_t^2\right), \tag{30}$$

*where*

$$\bar{G}_t = \frac{1}{H}\sum_{t'=t}^{t+H-1}\mathbb{E}_t[\|\nabla f(\bar{\mathbf{w}}_{t'})\|^2], \quad \bar{A}_t = \frac{1}{K}\sum_k \frac{1}{H}\sum_{t'=t}^{t+H-1}\mathbb{E}_t[\|\nabla f(\mathbf{w}_{t'}^k)\|^2]. \tag{31}$$

*Proof.* From $L$-smoothness, we have for any $t'$

$$\mathbb{E}_{t'}[f(\bar{\mathbf{w}}_{t'+1})] \leq f(\bar{\mathbf{w}}_{t'}) - \frac{\eta}{L}\nabla f(\bar{\mathbf{w}}_{t'})^T\mathbb{E}_{t'}[\bar{\mathbf{g}}_{t'}] + \frac{\eta^2}{2L}\mathbb{E}_{t'}[\|\bar{\mathbf{g}}_{t'}\|^2]$$

$$= f(\bar{\mathbf{w}}_{t'}) - \frac{\eta}{L}\nabla f(\bar{\mathbf{w}}_{t'})^T\left(\frac{1}{K}\sum_k \nabla f(\mathbf{w}_{t'}^k)\right) + \frac{\eta^2}{2L}\mathbb{E}_{t'}[\|\bar{\mathbf{g}}_{t'}\|^2] \tag{32}$$

Regarding the linear term, we have

$$\nabla f(\bar{\mathbf{w}}_{t'})^T \left( \frac{1}{K} \sum_k \nabla f(\mathbf{w}_{t'}^k) \right) = \frac{1}{2} \left( \frac{1}{K} \sum_k \|\nabla f(\mathbf{w}_{t'}^k)\|^2 + \|\nabla f(\bar{\mathbf{w}}_{t'})\|^2 - \frac{1}{K} \sum_k \|\nabla f(\mathbf{w}_{t'}^k) - \nabla f(\bar{\mathbf{w}}_{t'})\|^2 \right)$$

$$\geq \frac{1}{2} \left( \underbrace{\frac{1}{K} \sum_k \|\nabla f(\mathbf{w}_{t'}^k)\|^2 + \|\nabla f(\bar{\mathbf{w}}_{t'})\|^2}_{=:A_{t'}} - L^2 \underbrace{\frac{1}{K} \sum_k \|\mathbf{w}_{t'}^k - \bar{\mathbf{w}}_{t'}\|^2}_{=:V_{t'}} \right). \tag{33}$$

Furthermore, we have

$$\mathbb{E}_{t'}[\|\bar{\mathbf{g}}_{t'}\|^2] = \|\mathbb{E}_{t'}[\bar{\mathbf{g}}_{t'}]\|^2 + \mathbf{Var}_{t'}[\bar{\mathbf{g}}_{t'}] \leq \underbrace{\left\| \frac{1}{K} \sum_k \nabla f(\mathbf{w}_{t'}^k) \right\|^2}_{\leq A_{t'}} + \frac{\bar{\sigma}_t^2}{K}. \tag{34}$$

Plugging Equations (33) and (34) back into Equation (32), we get

$$\mathbb{E}_{t'}[f(\bar{\mathbf{w}}_{t'+1})] \leq f(\bar{\mathbf{w}}_{t'}) - \frac{\eta}{2L}\|\nabla f(\bar{\mathbf{w}}_{t'})\|^2 - \frac{\eta}{2L}A_{t'} + \frac{L\eta}{2}V_{t'} + \frac{\eta^2}{2L}A_{t'} + \frac{\eta^2}{2L}\frac{\bar{\sigma}_t^2}{K}. \tag{35}$$

Iterating this bound backward from $t' = t + H - 1$ while taking the expectation $\mathbb{E}_t$ yields

$$\mathbb{E}_t[f(\bar{\mathbf{w}}_{t+H})] = \mathbb{E}_t[\mathbb{E}_{t+H-1}[f(\bar{\mathbf{w}}_{t+H})]]$$

$$\leq \mathbb{E}_t \left[ f(\bar{\mathbf{w}}_{t+H-1}) - \frac{\eta}{2L}\|\nabla f(\bar{\mathbf{w}}_{t+H-1})\|^2 - \frac{\eta}{2L}A_{t+H-1} + \frac{L\eta}{2}V_{t+H-1} + \frac{\eta^2}{2L}A_{t+H-1} + \frac{\eta^2}{2L}\frac{\bar{\sigma}_t^2}{K} \right]$$

$$\leq \dots$$

$$\leq f(\bar{\mathbf{w}}_t) - \frac{\eta}{2L}\underbrace{\sum_{t'=t}^{t+H-1}\mathbb{E}_t[\|\nabla f(\bar{\mathbf{w}}_{t'})\|^2]}_{=H\bar{G}_t} - \frac{\eta}{2L}\underbrace{\sum_{t'=t}^{t+H-1}\mathbb{E}_t[A_{t'}]}_{=H\bar{A}_t} + \frac{\eta^2}{2L}\underbrace{\sum_{t'=t}^{t+H-1}\mathbb{E}_t[A_{t'}]}_{=H\bar{A}_t} + \frac{\eta^2}{2L}H\frac{\bar{\sigma}_t^2}{K} + \frac{L\eta}{2}\underbrace{\sum_{t'=t}^{t+H-1}\mathbb{E}_t[V_{t'}]}_{(*)} \tag{36}$$

It remains to bound the term $(*)$. From Lemma C.1 we know that $V_t$ obeys the recursion

$$\mathbb{E}_t[V_{t'}] \leq \mathbb{E}[V_{t'-1}] + \gamma^2\bar{\sigma}_t^2, \quad \gamma = \frac{\eta}{L} \tag{37}$$

Since $V_t = 0$, we get $\mathbb{E}_t[V_{t'}] \leq (t' - t)\bar{\sigma}_t^2\frac{\eta^2}{L^2}$ and

$$\sum_{t'=t}^{t+H-1}\mathbb{E}_t[V_{t'}] \leq \bar{\sigma}_t^2\eta^2\frac{1}{L^2}\sum_{t'=t}^{t+H-1}(t' - t) = \bar{\sigma}_t^2\eta^2\frac{(H-1)H}{2L^2}. \tag{38}$$

Plugging that back in gives

$$\mathbb{E}_t[f(\bar{\mathbf{w}}_{t+H})] \leq f(\bar{\mathbf{w}}_t) - \frac{\eta}{2L}H\bar{G}_t - \frac{\eta}{2L}H\bar{A}_t + \frac{\eta^2}{2L}H\bar{A}_t + \frac{\eta^2}{2L}H\frac{\bar{\sigma}_t^2}{K} + \frac{\eta^3}{4L}(H-1)H\bar{\sigma}_t^2 \tag{39}$$

$\square$

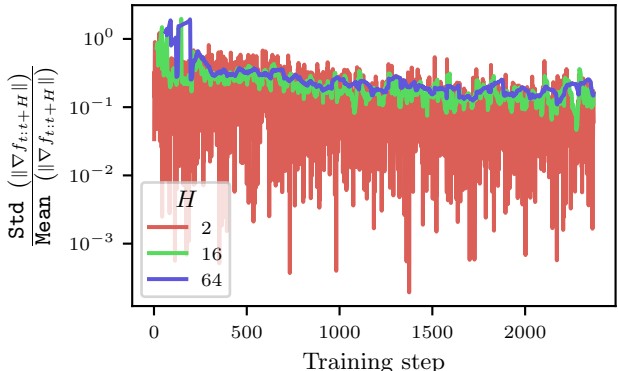

Figure 7: Examining the assumption in Equation (40). We plot the standard deviation of the full-gradient magnitude over $H$ steps for a ResNet-9 (He et al., 2016) trained on CIFAR-10 (Krizhevsky, 2009). It is evident that the gradient magnitude changes very little over the course of training and thus the assumption over a small number of $H$ gradient steps the magnitude remains constant is a reasonable approximation.

This expected function decrease bound features terms $\bar{G}_t$ and $\bar{A}_t$, which are expected squared gradient magnitudes along the trajectories of the virtual averaged iterates, and the per-worker iterates, respectively. These quantities are not computed in practice, and in principle, dependent on $\eta$, since the choice of step size influences the gradient magnitude along that trajectory. We make a simplifying assumption that

$$\bar{A}_t \approx \|\nabla f(\bar{\mathbf{w}}_t)\|^2 \approx \bar{G}_t \tag{40}$$

Equation (40) signifies that between two synchronizations the average virtual gradient magnitude and the local gradient magnitudes computed on each worker do not change too much and can be approximated by the gradient magnitude computed at the point of synchronization ($\bar{\mathbf{w}}_t$). We study the validity of this assumption in Figure 7. We train a small ResNet-9 on CIFAR-10 using SGD. We are interested in how the gradient magnitude (not stochastic gradient magnitude) evolves over a short horizon of $H$ steps, and thus we plot the ratio standard deviation of the gradient magnitude over $H$ steps to the average gradient magnitude over those $H$ steps, commonly termed coefficient of variation. It is apparent from Figure 7 that the coefficient of variation is well below is almost $< 1$ except at the start of the training. This provides evidence that we can reuse the gradient magnitude for $H$ steps due to its relative constancy.

Substituting Equation (40) into Equation (39) gives us

$$\mathbb{E}_t[f(\bar{\mathbf{w}}_{t+H})] \leq f(\bar{\mathbf{w}}_t) - H\left(\gamma \bar{G}_t - \frac{\gamma^2 L}{2}\bar{G}_t - \frac{\gamma^2 L}{2}\frac{\bar{\sigma}_t^2}{K} - \frac{\gamma^3 L^2}{4}(H-1)\bar{\sigma}_t^2\right) \tag{41}$$

### C.3 Optimal Step Size (Equation 13)

We can now find the learning rate $\eta$ that minimizes our bound on $\mathbb{E}[f(\bar{\mathbf{w}}_{t+H})]$ in Equation (41). Setting the derivative of the right-hand side w.r.t. $\gamma$ to zero reads

$$\bar{G}_t - \gamma L \bar{G}_t - \gamma L \frac{\bar{\sigma}_t^2}{K} - \frac{3}{4}\gamma^2 L^2(H-1)\bar{\sigma}_t^2 = 0 \tag{42}$$

This quadratic equation in $\gamma$ can be solved by the standard quadratic formula. We instead use Muller's method (Muller, 1956) which gives the roots of a quadratic of form $ax^2 + bx + c = 0$ as

$$x = \frac{2c}{-b \pm \sqrt{b^2 - 4ac}} \tag{43}$$

Ignoring the negative root, we get

$$\gamma = \frac{1}{L} \frac{2\bar{G}_t}{\bar{G}_t + \bar{\sigma}_t^2/K + \sqrt{(\bar{G}_t + \bar{\sigma}_t^2/K)^2 + 3(H-1)\bar{G}_t\bar{\sigma}_t^2}}. \tag{44}$$

In our experiments, following Johnson et al. (2020), we use SGD with momentum. Momentum buffers are maintained locally on each machine and not synchronized. We leave the investigation of different approaches to momentum buffers (Lin et al., 2020; Wang et al., 2020; Chen & Huo, 2016) to future work.

## D  Expected Decrease for $H$ Steps of Synchronous SGD

To elucidate the expected decrease bound for Local SGD in Equation (41), we contrast it with a similar bound for $H$ consecutive steps of synchronous data-parallel SGD using a constant learning rate $\gamma$. Analogously to our analysis for Local SGD, we assume gradient variance and magnitude to be bounded across these $H$ steps,

$$\mathbb{E}_{t'}[g_{t'}^k] \leq \bar{\sigma}_t^2, \quad \|\nabla f(\mathbf{w}_{t'})\|^2 \geq \bar{G}_t, \quad t' \in [t, t+H). \tag{45}$$

For the averaged gradient across $K$ workers, this results in $\mathbf{Var}_{t'}[\tilde{\mathbf{g}}_{t'}] \leq \frac{\bar{\sigma}_t^2}{K}$.

By smoothness, we get

$$
\begin{aligned}
\mathbb{E}_{t'}[f(\mathbf{w}_{t'+1})] &\leq f(\mathbf{w}_{t'}) - \gamma \nabla f(\mathbf{w}_{t'})^T \mathbb{E}_{t'}[\tilde{\mathbf{g}}_{t'}] + \frac{L\gamma^2}{2}\mathbb{E}_{t'}\left[\|\tilde{\mathbf{g}}_{t'}\|^2\right] \\
&= f(\mathbf{w}_{t'}) - \left(\gamma\|\nabla f(\mathbf{w}_{t'})\|^2 - \frac{L\gamma^2}{2}\left[\|\nabla f(\mathbf{w}_{t'})\|^2 + \mathbf{Var}_{t'}[\tilde{\mathbf{g}}_{t'}]\right]\right) \\
&\leq f(\mathbf{w}_{t'}) - \left(\gamma\bar{G}_t - \frac{L\gamma^2}{2}\left[\bar{G}_t + \frac{\bar{\sigma}_t^2}{K}\right]\right).
\end{aligned}
\tag{46}
$$

Using this recursively for $H$ steps from $t$ to $t+H$ while taking the expectation $\mathbb{E}_t$, we get

$$\mathbb{E}_t[f(\mathbf{w}_{t+H})] \leq f(\mathbf{w}_t) - \underbrace{H \cdot \left(\gamma\bar{G}_t - \frac{\gamma^2 L}{2}\bar{G}_t - \frac{\gamma^2 L}{2}\frac{\bar{\sigma}_t^2}{K}\right)}_{\text{Expected decrease for SGD}}. \tag{47}$$

Contrast this with Equation (30), restated here:

$$\mathbb{E}_t[f(\bar{\mathbf{w}}_{t+H})] \leq f(\bar{\mathbf{w}}_t) - H \cdot \left(\gamma\bar{G}_t - \frac{\gamma^2 L}{2}\bar{G}_t - \frac{\gamma^2 L}{2}\frac{\bar{\sigma}_t^2}{K} - \frac{\gamma^3 L^2}{4}(H-1)\bar{\sigma}_t^2\right). \tag{48}$$

We see that Local SGD has an additional cubic term that results from the biased estimation of gradients.

## E  Avoiding Stale Gradients in Computing the Gain Ratio

To implement LocalAdaScale, we need to estimate $\bar{G}_t$ and $\bar{\sigma}_t^2$. In Algorithm 1, we proposed a method that synchronizes *once* every $H$ steps to average the weights and the cached stale gradients simultaneously. As an ablation study, we compare this against an alternative strategy where we synchronize the gradients one step after weight synchronization in Algorithm 4. While this incurs latency overhead *twice*, the resulting gain ratio is computed with more recent gradient evaluations and should therefore be more accurate.

In Figure 9, we compare the two variants Local-Ada, and Local-Ada-NoStale. Across all datasets, we find that there are very small differences between the two variants in the final accuracy obtained or the number of epochs to convergence. The difference in the number of epochs for Local-Ada and Local-Ada-NoStale is explained by the fact that underestimating the gain ratio $\rho$ due to using stale gradients results in longer training durations.

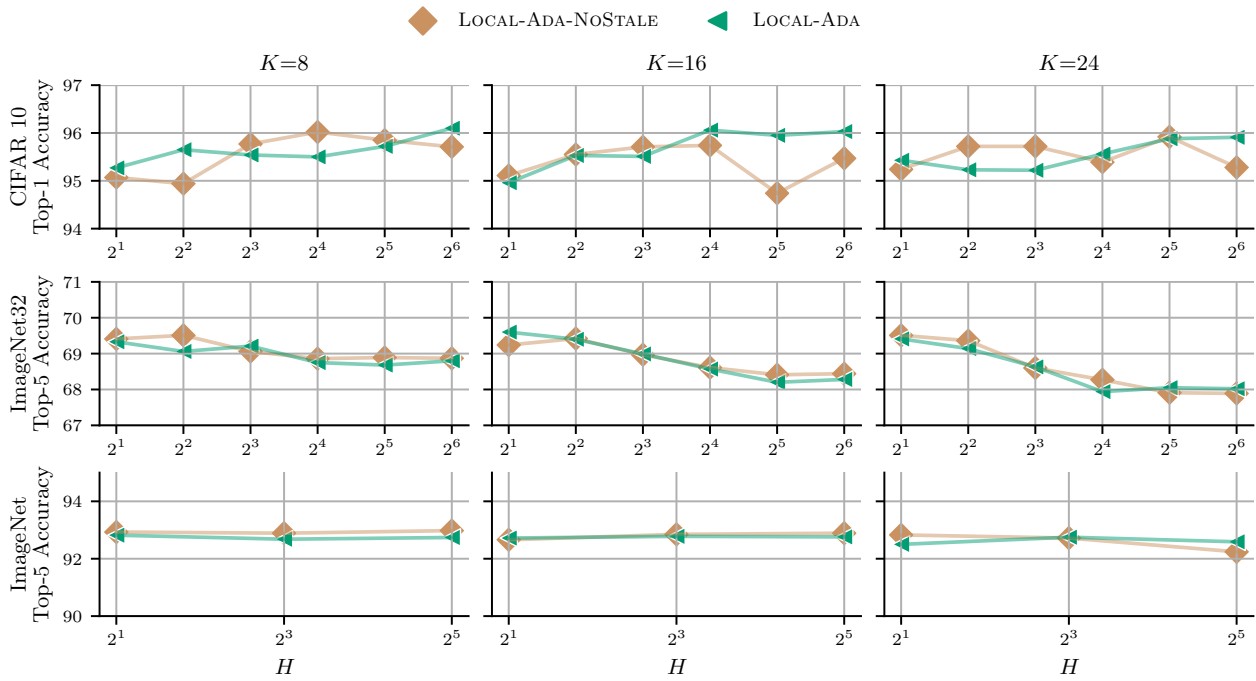

Figure 8: Comparison of test accuracies achieved for different numbers of workers ($K$) and communication intervals ($H$) by the two implementation variants of LocalAdaScale. Accuracy stays nearly identical between the two implementations, and the minor variations are within random seed variations.

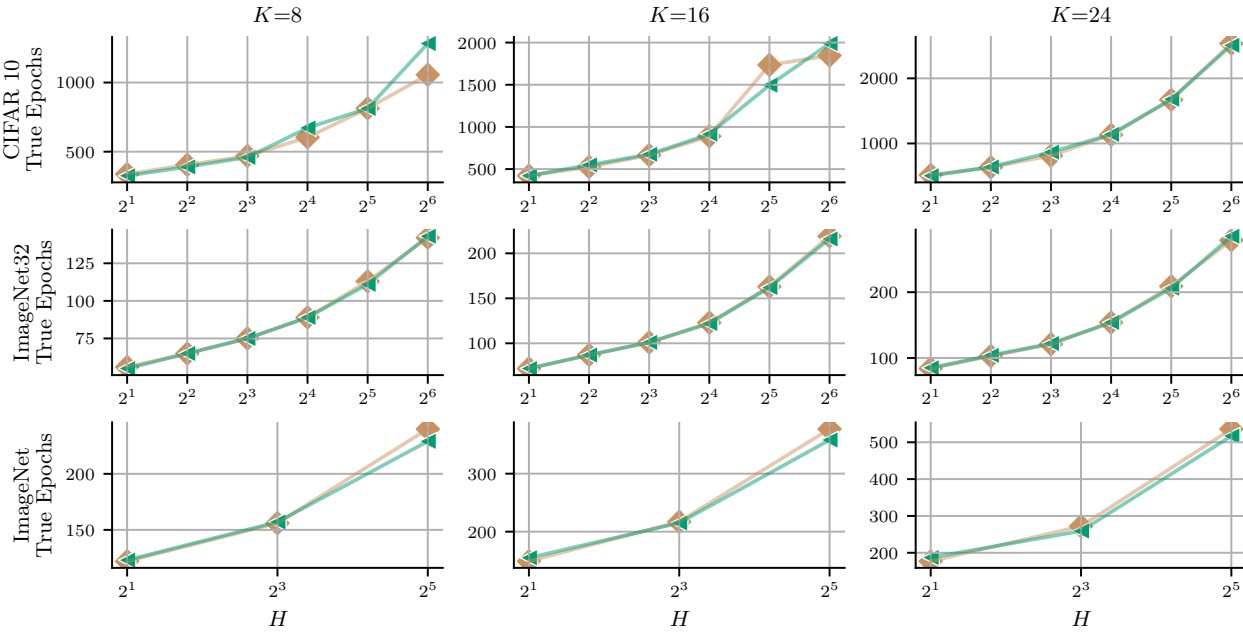

Figure 9: Number of epochs used by each method. When $K > 8$, both variants converge in a nearly identical number of epochs. LOCAL-ADA trains for marginally more iterations, as it underestimates the gain ratio due to the utilization of cached gradients

---

**Algorithm 4** LocalAdaScale with two synchronizations – LOCAL-ADA-NOSTALE

---
1: Input: Initialization $\mathbf{w}_0$, step-size $\gamma_t$, #workers $K$, #local steps $H$, Scale inv budget $\mathcal{S}$, $t = 0$, $s = 0$.
2: **while** $s \leq \mathcal{S}$ **do**                                                                      ▷ Scale inv budget not exhausted.
3:     **for** $k \in [K]$ **do**                                                                              ▷ On each worker
4:         Compute $\mathbf{g}_t^k$ using a batch of data.                                                     ▷ Gradient at $t$.
5:         **if** $H \mid t$ **then**                                                                          ▷ One step after model sync.
6:             $\bar{G}_t, \bar{\sigma}_t^2 \leftarrow \text{grad\_stats}(\mathbf{g}_t^k)$                      ▷ Equation (15)
7:             Compute $\rho$ as Equation (14).
8:         **if** $H \mid (t+1)$ **then**                                                                      ▷ Average model every $H$ steps
9:             $\mathbf{w}_t^j \leftarrow \frac{1}{K} \sum_{j=1}^{K} \mathbf{w}_t^j \quad \forall j.$
10:        $\mathbf{w}_{t+1}^k \leftarrow \mathbf{w}_t^k - \rho \gamma_{\lceil s \rceil} \mathbf{g}_t^k.$       ▷ Local update.
11:        **if** $H \mid (t+1)$ **then**                                                                      ▷ Average every $H$ steps
12:            $\mathbf{w}_t^j \leftarrow \frac{1}{K} \sum_{j=1}^{K} \mathbf{w}_t^j \quad \forall j.$
13:        $s \leftarrow s + \rho_t$
14:        $t \leftarrow t + 1$
15: **return** the last iterate $\mathbf{w}_i$.

---

## F   How accurate are our approximations in Equation (14)?

We study the tightness of our assumptions by studying a generalized version of Equation (14) as

$$\rho_t(\boxed{c}) = \frac{2\left(\bar{G}_t + \bar{\sigma}_t^2\right)}{\bar{G}_t + \frac{\bar{\sigma}_t^2}{K} + \sqrt{\left(\bar{G}_t + \frac{\bar{\sigma}_t^2}{K}\right)^2 + 3\,\boxed{c}\,(H-1)\bar{G}_t\bar{\sigma}_t^2}}. \tag{49}$$

Here when $\rho_t(0)$ reduces to AdaScale and $\rho_t(1)$ is LocalAdaScale, and thus $\rho_t(c)$ for $c \in [0, 1]$ interpolates AdaScale and LocalAdaScale. Modulating $c$ scales the learning rate as well as the number of training iterations.

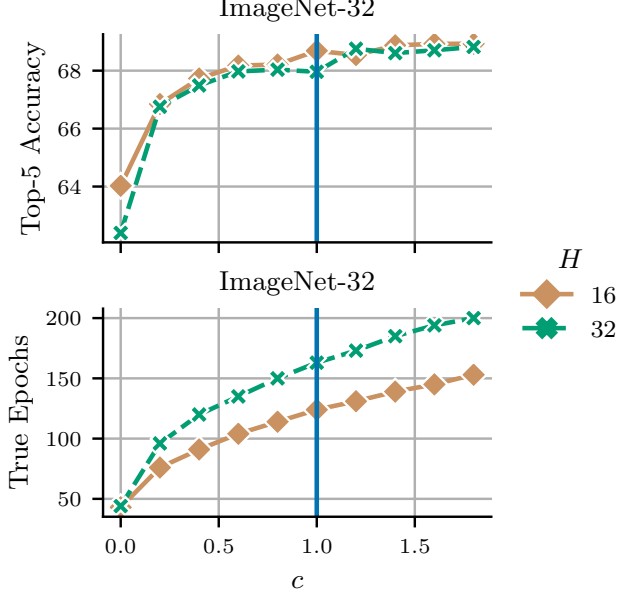

Figure 10: Ablation over $c$ in Equation (49). We plot $c$ *vs* Top-5 Accuracy and Number of epochs ImageNet32 for $K = 16$ for $H = 16, 32$. We see that $c \geq 0.6$ results recovering the Target Accuracy.

We show in Figure 10 the effect of $c$ on the performance. Using AdaScale's gain formula ($c = 0$) for Local SGD is has the advantage of fastest convergence albeit with poorer generalization as the learning rate used is likely too aggressive, and consequently training iterations too few. As $c$ is increased from 0, we see improved performance and the performance shows diminishing returns when $c \geq 0.6$. This threshold can vary with the dataset and network architecture in addition to the number of workers, and cannot be interpreted as a universal threshold. We see that our proposed method scales the learning rate conservatively and our threshold works well for very high communication gap of $H = 32$. This figure is further evidence that scaling methods that were designed for synchronous data-parallel SGD do not perform well for Local SGD and the delayed communication ($H \geq 2$) has to be accounted for, as we seen in the case of $c = 0$ where there is a substantial performance difference between the two cases $H = 16$ and $H = 32$.

## G  Need to Modulate Step Size for Larger Batch Sizes

### G.1  With Constant Learning Rate

Khaled et al. (2020); Stich (2019) analyze the convergence of Local SGD under a different condition for learning rate: either constant, or decreasing with time, but not the optimal learning rate. We can analyze this choice by mapping that idea to SGD in Equation (18). Let $\gamma$ be the learning rate that is small enough (additional upper bounds exist). We can see that with that learning rate the loss function decreases by

$$\mathbb{E}[f(\mathbf{w}_{t+1})|\mathbf{w}_t] \leq f(\mathbf{w}_t) - \left( \gamma \|\nabla f_t\|^2 - \frac{\|\nabla f_t\|^2 + \sigma_t^2}{2} L\gamma^2 \right) \tag{50}$$

By keeping the learning rate the same value and when the number of workers goes up $K$ fold, the variance reduces by the same factor, we get

$$\mathbb{E}[f(\mathbf{w}_{t+1})|\mathbf{w}_t] \leq f(\mathbf{w}_t) - \left( \gamma \|\nabla f_t\|^2 - \frac{\|\nabla f_t\|^2 + \frac{\sigma_t^2}{K}}{2} L\gamma^2 \right) \tag{51}$$

Comparing Equations (50) and (51), it is apparent that merely increasing the batch size results in an update that results in a larger reduction of the function value without having to tinker with the learning rate at all. However, this is suboptimal as a practitioner would be interested in tuning the learning rate that results in the largest reduction (implying faster training). Here lies the difference between prior works and our results.

### G.2  Using AdaScale

In Appendix B, we derived the optimal learning rate, which results in the largest decrease in the function value. Thus, when the number of workers is increased from 1 to $K$, any learning rate smaller than the optimal, will also result in the decrease of the function value in Equation (18), however, will be lesser than optimal. We make this more formal here.

Substituting Equation (19) into Equation (18), we get

$$\mathbb{E}[f(\mathbf{w}_{t+1})|\mathbf{w}_t] \leq f(\mathbf{w}_t) - \left( \frac{\|\nabla f_t\|^2}{(\sigma_t^2 + \|\nabla f_t\|^2)} \frac{\|\nabla f_t\|^2}{L} - \frac{(\sigma_t^2 + \|\nabla f_t\|^2)}{2L} \frac{\|\nabla f_t\|^4}{(\sigma_t^2 + \|\nabla f_t\|^2)^2} \right)$$

$$= f(\mathbf{w}_t) - \left( \frac{1}{2L} \underbrace{\frac{\|\nabla f_t\|^4}{(\sigma_t^2 + \|\nabla f_t\|^2)}}_{\text{Reduction factor}} \right) \qquad \text{(OptRednSGD-1 Worker)}$$

Similarly, we can show that the optimal reduction in the function value when $K$ workers are used as

$$\mathbb{E}[f(\mathbf{w}_{t+1})|\mathbf{w}_t] \leq f(\mathbf{w}_t) - \left( \frac{1}{2L} \underbrace{\frac{||\nabla f_t||^4}{\left( \frac{\sigma_t^2}{K} + ||\nabla f_t||^2 \right)}}_{\text{Reduction factor}} \right) \qquad \text{(OptRednSGD-K Worker)}$$

The reduction factor in Equation (OptRednSGD-K Worker) $\geq$ Equation (OptRednSGD-1 Worker), as the denominator in Equation (OptRednSGD-K Worker) is smaller than Equation (OptRednSGD-1 Worker) and all other terms are identical. A learning rate adaptation is required to capitalize on the availability of $K$ workers.

## H  Details of Experiments

### H.1  Experimental Setup

**CIFAR-10**  We used ResNet model code and training hyperparameters from the GitHub repository `https://github.com/kuangliu/pytorch-cifar`. They report a test accuracy of 93.02% for ResNet-18 in the single-worker base setting.

**ImageNet32**  We used Wide ResNet model code from `https://github.com/weiaicunzai/pytorch-cifar100` and training hyperparameters from the paper proposing ImageNet32 (Chrabaszcz et al., 2017). They report a top-5 test accuracy of 69.08% for WRN-28-2 in the single-worker base setting.

**ImageNet**  We used ResNet model from the torchvision library and training hyperparameters from Goyal et al. (2018). The torchvision library reports a top-5 test accuracy of 92.86%.

Training hyperparameters are listed in the following table:

| Dataset | $\gamma_{\text{base}}$ | Momentum | Weight decay | LR schedule | Epochs |
|---|---|---|---|---|---|
| CIFAR-10 | 0.1 | 0.9 | $5 \times 10^{-4}$ | Cosine-decay | 200 |
| ImageNet32 | 0.01 | 0.9 | $5 \times 10^{-4}$ | Step ($\times 0.5$ every 10 epochs) | 40 |
| ImageNet | 0.1 | 0.9 | $5 \times 10^{-4}$ | Step ($\times 0.1$ every 30 epochs) | 90 |

### H.2  Tabulated Results

Tabulated results of our experiments may be found in Tables 2 to 4.

Table 2: Accuracy [%] and number of epochs for CIFAR-10.

| $K$ | $H$ | Accuracy | | | | Epochs | | | |
|---|---|---|---|---|---|---|---|---|---|
| | | Acc-Ada | Acc-Lin | Local-Ada | Local-Lin | Acc-Ada | Acc-Lin | Local-Ada | Local-Lin |
| 8 | 2 | 95.12 | 94.73 | 95.07 | 95.16 | 291 | 200 | 338 | 200 |
| | 4 | 95.11 | 93.92 | 94.94 | 95.26 | 353 | 200 | 405 | 200 |
| | 8 | 94.81 | 88.06 | 95.77 | 95.22 | 496 | 200 | 469 | 200 |
| | 16 | 94.70 | 53.09 | 96.02 | 94.98 | 791 | 200 | 604 | 200 |
| | 32 | 94.28 | 13.19 | 95.85 | 94.25 | 1450 | 200 | 813 | 200 |
| | 64 | 94.53 | 10.70 | 95.71 | 93.61 | 2260 | 201 | 1056 | 200 |
| 16 | 2 | 95.35 | 94.13 | 95.11 | 95.24 | 360 | 200 | 422 | 200 |
| | 4 | 95.15 | 87.36 | 95.55 | 94.69 | 502 | 200 | 525 | 200 |
| | 8 | 94.65 | 36.52 | 95.71 | 93.82 | 812 | 200 | 666 | 200 |
| | 16 | 94.43 | 14.18 | 95.74 | 93.35 | 1330 | 200 | 890 | 200 |
| | 32 | 94.64 | 14.12 | 94.74 | 91.95 | 2499 | 201 | 1735 | 200 |
| | 64 | 93.24 | 10.12 | 95.47 | 90.83 | 6157 | 202 | 1846 | 200 |
| 24 | 2 | 94.52 | 90.02 | 95.24 | 94.21 | 443 | 200 | 515 | 200 |
| | 4 | 94.79 | 79.58 | 95.72 | 93.63 | 617 | 200 | 634 | 200 |
| | 8 | 94.55 | 13.87 | 95.72 | 92.99 | 1006 | 200 | 813 | 200 |
| | 16 | 93.65 | 10.76 | 95.39 | 91.68 | 1634 | 200 | 1132 | 200 |
| | 32 | 93.96 | 11.08 | 95.92 | 86.69 | 3688 | 201 | 1671 | 200 |
| | 64 | 92.73 | 11.48 | 95.28 | 66.22 | 8542 | 203 | 2535 | 200 |

Table 3: Accuracy [%] and number of epochs for ImageNet32.

| $K$ | $H$ | Accuracy | | | | Epochs | | | |
|---|---|---|---|---|---|---|---|---|---|
| | | Acc-Ada | Acc-Lin | Local-Ada | Local-Lin | Acc-Ada | Acc-Lin | Local-Ada | Local-Lin |
| 8 | 2 | 69.01 | 68.00 | 69.41 | 68.32 | 49 | 40 | 56 | 40 |
| | 4 | 69.15 | 67.84 | 69.51 | 68.13 | 57 | 40 | 65 | 40 |
| | 8 | 69.48 | 62.59 | 69.06 | 67.31 | 74 | 40 | 75 | 40 |
| | 16 | 69.50 | 51.15 | 68.86 | 66.19 | 104 | 40 | 89 | 40 |
| | 32 | 69.91 | 41.72 | 68.89 | 64.76 | 161 | 40 | 113 | 40 |
| | 64 | 69.94 | 28.45 | 68.87 | 63.96 | 283 | 40 | 142 | 40 |
| 16 | 2 | 68.88 | 67.35 | 69.24 | 68.27 | 58 | 40 | 72 | 40 |
| | 4 | 69.16 | 61.09 | 69.42 | 67.22 | 73 | 40 | 87 | 40 |
| | 8 | 69.66 | 53.42 | 68.97 | 65.80 | 104 | 40 | 101 | 40 |
| | 16 | 70.01 | 48.60 | 68.61 | 63.44 | 162 | 40 | 123 | 40 |
| | 32 | 70.02 | 29.78 | 68.41 | 60.93 | 286 | 40 | 163 | 40 |
| | 64 | 69.54 | 6.30 | 68.44 | 59.27 | 518 | 40 | 219 | 40 |
| 24 | 2 | 68.99 | 66.51 | 69.51 | 67.02 | 65 | 40 | 84 | 40 |
| | 4 | 69.45 | 45.41 | 69.36 | 65.81 | 87 | 40 | 103 | 40 |
| | 8 | 69.71 | 47.79 | 68.60 | 64.05 | 133 | 40 | 121 | 40 |
| | 16 | 69.58 | 32.45 | 68.27 | 60.80 | 218 | 40 | 154 | 40 |
| | 32 | 69.77 | 7.91 | 67.91 | 56.53 | 409 | 40 | 209 | 40 |
| | 64 | 68.97 | 4.84 | 67.89 | 54.18 | 744 | 40 | 279 | 40 |

Table 4: Accuracy [%] and number of epochs for ImageNet.

| $K$ | $H$ | Accuracy | | | | Epochs | | | |
|---|---|---|---|---|---|---|---|---|---|
| | | Acc-Ada | Acc-Lin | Local-Ada | Local-Lin | Acc-Ada | Acc-Lin | Local-Ada | Local-Lin |
| 8 | 2 | 92.32 | 92.66 | 92.93 | 92.31 | 108 | 90 | 122 | 90 |
| | 8 | 92.54 | 90.70 | 92.89 | 92.30 | 157 | 90 | 156 | 90 |
| | 32 | 92.14 | 59.23 | 92.98 | 90.72 | 328 | 90 | 240 | 90 |
| 16 | 2 | 92.15 | 92.37 | 92.66 | 92.54 | 127 | 90 | 149 | 90 |
| | 8 | 92.54 | 85.40 | 92.85 | 91.64 | 213 | 90 | 217 | 90 |
| | 32 | 91.74 | 0.58 | 92.89 | 88.40 | 568 | 90 | 377 | 90 |
| 24 | 2 | 92.84 | 91.81 | 92.83 | 92.49 | 141 | 90 | 177 | 90 |
| | 8 | 92.35 | 77.93 | 92.72 | 90.83 | 260 | 90 | 272 | 90 |
| | 32 | 91.12 | 0.62 | 92.24 | 84.08 | 761 | 90 | 536 | 90 |

## H.3 Additional Results for Pseudo-Wall Clock Time

In Figure 11, we plot the full communication tradeoffs for all the datasets, and workers ($K$) considered.

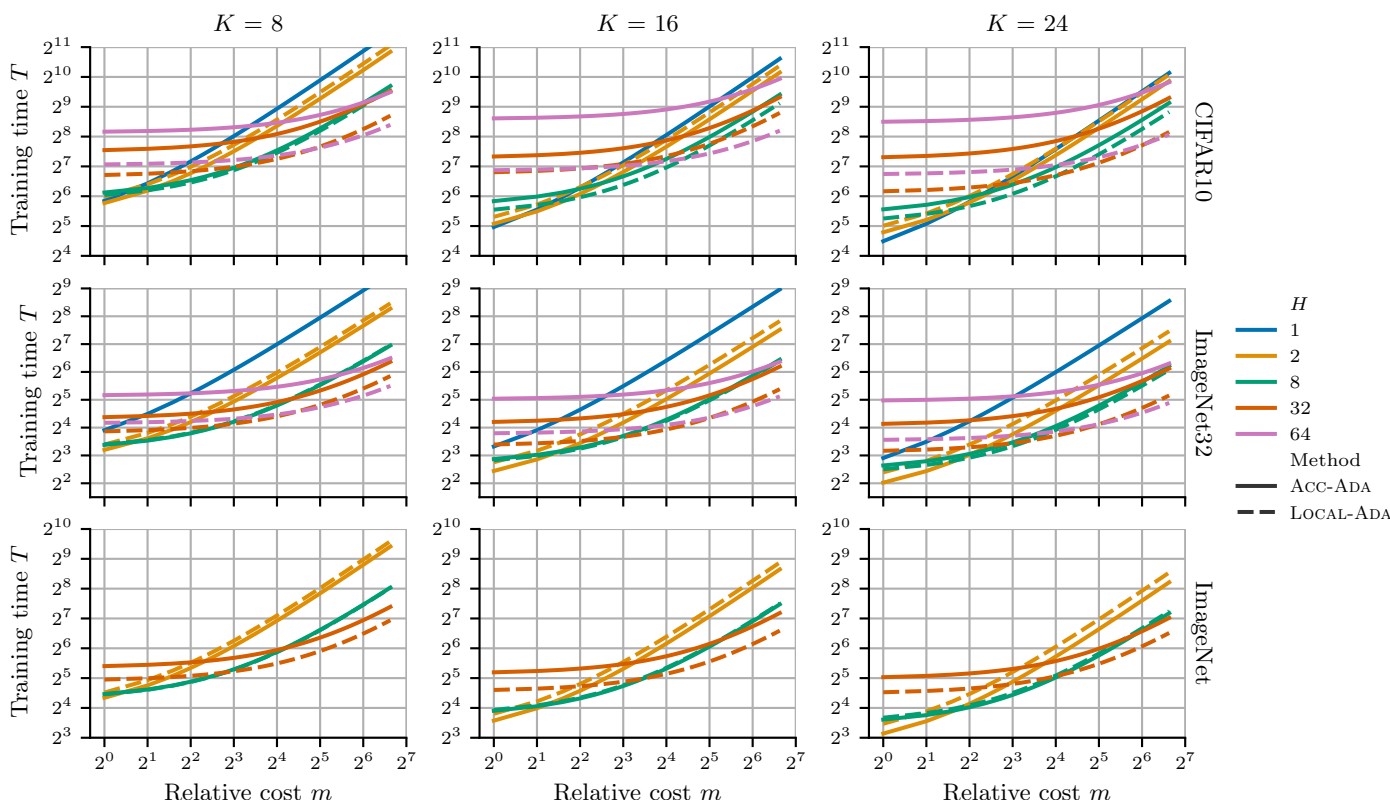

Figure 11: When is LocalAdaScale preferable to Acc-Ada? On the x-axis, we plot the relative cost $m$ of communication to computation, and on the y-axis, we plot pseudo-wallclock time $T$. We see that LocalAdaScale converges faster than gradient accumulation for higher $H$, and also for a higher cost of communication. For lower $m$, gradient accumulation with fewer steps is preferable to LocalAdaScale.

## H.4 Behavior of the Gain Ratio

Figures 12 and 13 show the gain ratios used by AdaScale and LocalAdaScale. We plot $\rho$ (Eq. 14) for Local-Ada and the "effective gain" $\frac{r}{H}$ (Eq. 9) for Acc-Ada, accounting for the different scales of $\rho$ and $r$. Since different methods take different numbers of iterations, we adjust the $x$-axis to correspond to scale-invariant epochs. We see that both methods approach the maximum gain ratio of $K$ for small values of $H$. The average gain ratio achieved by Acc-Ada is slightly higher for small $H$, resulting in fewer iterations as observed before. For large $H$, this is reversed, and Local-Ada achieves higher average gains. In all settings and for both methods, the gain ratio generally increases smoothly over time, reflecting a diminishing gradient magnitude.

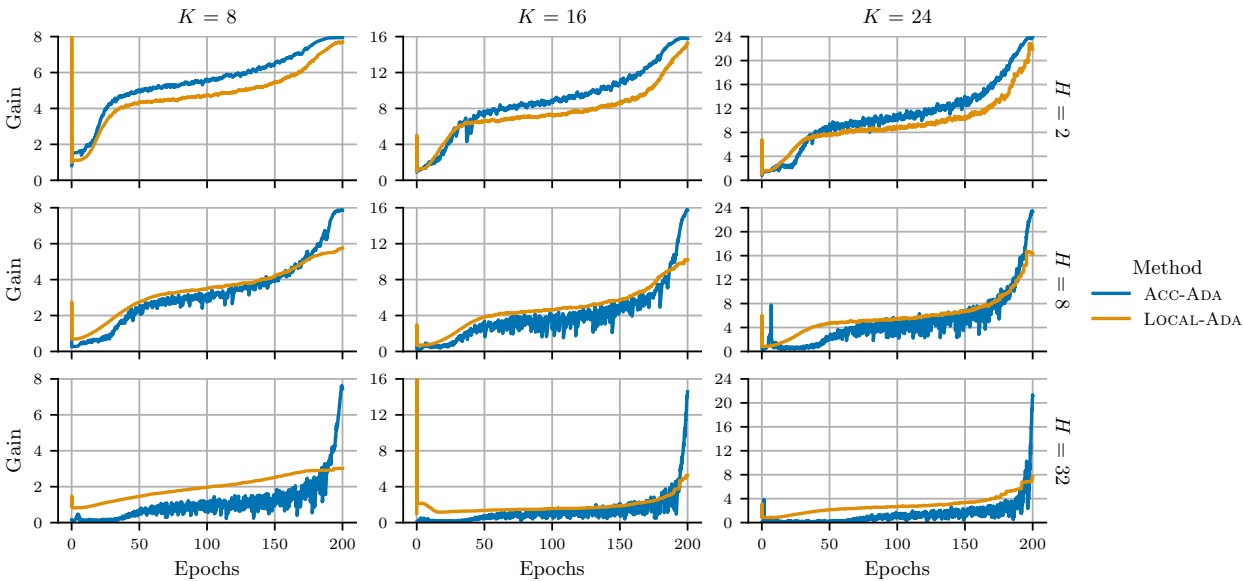

Figure 12: Gain ratios for CIFAR-10 on ResNet-18. Each row corresponds to a communication interval $H$ and each column to the number of workers $K$. ACC-ADA reaches higher gain ratios for small $H$ and thus is more iteration efficient than LOCAL-ADA. This trend reverses for large $H$, where we see that ACC-ADA reaches higher scaling only towards the end of training. The x-axis has been linear downsampled to fit into 200 epochs for visualization.

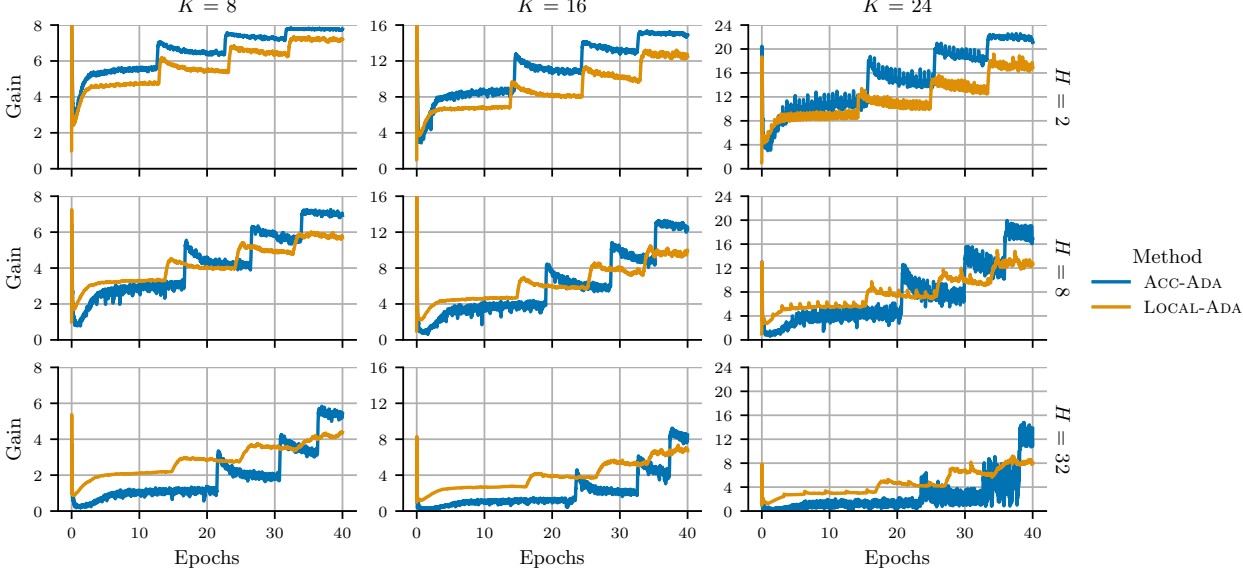

Figure 13: Gain ratios for ImageNet-32 on WRN-28-2. Each row corresponds to a communication interval $H$ and each column to the number of workers $K$. ACC-ADA reaches higher gain ratios for small $H$ and thus is more iteration efficient than LOCAL-ADA. This trend reverses for large $H$, where we see that ACC-ADA reaches higher scaling only towards the end of training. The x-axis has been linear downsampled to fit into 40 epochs for visualization. The step structure of the gain is due to the step learning rate decay schedule and does not align across plots because LocalAdaScale uses scale invariant epochs, which corresponds to different number of epochs in each $K, H$ setting.

