# OpenReview forum: "On the Choice of Learning Rate for Local SGD"
_TMLR — Accepted by TMLR_

### Review · Reviewer_3XYf · 2023-10-16

**Summary Of Contributions:**

This article studies the importance of the learning rate choice in the local SGD method for distributed optimization. Compared to the data-parallel SGD method, there is a bias in the update direction of this method, which makes the method fail to converge when the learning rate is not suitably chosen. An adaptive method is proposed, based on existing works for data-parallel SGD. Extensive numerical results to show the advantage of using local SGD relative to the data-parallel SGD, when the commutation between workers is costly.

**Audience:**

Yes

**Broader Impact Concerns:**

The main result of the article remains still empirical, but the topic is interesting so it may open up new research directions in the future.

**Claims And Evidence:**

No

**Requested Changes:**

-	In the numerical results in section 6, what is the definition of each iteration (in order to count n^{K,H} in eq 16) for the gradient accumulation methods? Is it the same k as the local SGD method? It is important to be precise on this.
-	As mentioned in the beginning of section 6.1, one seeks to achieve a quite high test accuracy in each of the 3 dataset. Did you use this accuracy to decide the value of n^{K,H}? If so, how? Also why in Fig 5, the local-lin method with K=24 can still achieve such an accuracy (with a quite small number of epochs)? Is it contradictory to the result in Fig 4 ? Is the number of epochs  = n^{K,H} / training data size ?
-	There is still a lack of understanding why the local SGD with the proposed learning rate scheduling can achieve a good accuracy. Is it due to the reduced bias as mentioned in section 4? Some further discussions on this would be good.
-	In the abstract, the cost of added training iterations of local SGD is with respect to which method? Unlike in the synchronous case, which case are you referring to? In the introduction, when is local SGD empirically preferable to SGD, you mentioned local SGD converges faster than large batch SGD. What is the measurement of the speed here (to be faster)?

**Strengths And Weaknesses:**

The article gives a good literature review over the topic. The presentation is quite clear and easy to read. The idea of using adaptive learning rate as in AdaScale to reduce the bias in eq 6 seems to be new to the local SGD literature. It is nevertheless unclear how much bias is reduced in the proposed algorithm 1. Further clarifications are needed for me to understand the numerical results.

---

### Review · Reviewer_yXqE · 2023-10-25

**Summary Of Contributions:**

This work studies the convergence of Local SGD. In particular, the authors focus on ERM problems and derive an upper bounded on the expected decrease in the loss function in each iteration as a function of the number of workers $K$ and the number of local iterations $H$, when assuming L-smooth functions. They then derive a heuristic learning rate adaptation strategy that minimizes their upper bound on the expected training loss after $H$ local iterations. When $H=1$, they recover the AdaScale method, and thus they term their proposed strategy Local AdaScale. Following the AdaScale paper, the "effective" number of training iterations is tracked based on a specified "gain scale," which depends on the online gradient norms, an upper bound on the stochastic gradient variance, and the learning rate... coarsely speaking using a smaller learning rate can to a first order result in a slower iteration counter, and thus require more physical iterations to train for a given "effective" number of iterations. As a result, the proposed method not only adapts the Local SGD learning rate, but also the total number of Local SGD training iterations.

Numerical experiments are provided on supervised image classification with convolutional networks. The primary baselines are a) linear learning rate scaling with Synchronous SGD with $K$ workers and $H$ gradient accumulation steps, b) AdaScale learning rate scaling with Synchronous SGD with $K$ workers and $H$ gradient accumulation steps, c) linear learning rate scaling with Local SGD with $K$ workers and $H$ local gradient steps, d) Local AdaScale learning rate scaling with Local SGD with $K$ workers and $H$ local gradient steps.

Empirically, the proposed Local AdaScale strategy results in better generalization on the validation set than linear learning rate scaling with Local SGD. However, the authors also find that Local SGD requires a significant imbalance between communication and computation time (at least 3x in many cases) to provide any wall-clock speedups compared to large mini-batch SGD, since Local AdaScale can require many more training iterations to reach a certain level of performance.

**Audience:**

Yes

**Broader Impact Concerns:**

No concerns.

**Claims And Evidence:**

Yes

**Requested Changes:**

* Please include a discussion and clarify how to reconcile the theory derived with respect to the training loss, and the practical quantity of interest, generalization and validation accuracy.
* Please clarify and make it clear in the main paper how momentum is used in the numerical experiments. Are there both global and local momentum buffers? If so, how do they interact? If not, how do you synchronize momentum buffers? Would properly handling momentum buffers change the conclusions or decrease the performance gap? How would the results change if one incorporated existing techniques for correctly tracking the momentum buffers, as in SloMo?
* Please clarify whether the numerical experiments are conducted with a pre-specified step-wise learning rate decay rule on top of the proposed Local AdaScale rule. If so, please discuss how you reconcile this gap between theory and practice?
* Please include a broader discussion on learning rate adaptation methods and their use in the literature.

My main concern with this work is the large gap between theory (motivation and derivation of an adaptive learning rate), and practice (unsynchronized momentum buffers, step-wise decay, validation accuracy vs training loss, etc.). I am concerned that the simple and well written exposition may provide a misleading picture for researchers, students, or practitioners trying to understand the behaviour of Local SGD.

**Strengths And Weaknesses:**

### Strengths
* This work is very well written and easy to follow; indeed, it was a pleasure to read this paper.
* The theory is also quite straightforward and provides a clear motivation for the proposed adaptive learning rate scaling.
* Numerical experiments provide compelling evidence for the advantage of the proposed adaptive learning rate scaling method compared to the linear scaling heuristic.
* The paper is also quite honest in taking a critical look at when one would expect Local SGD to be advantageous in practice, and the authors are not afraid to point out that extreme communication imbalance is required in practice to realize a speedup with Local SGD.

### Weaknesses
* All arguments in the paper follow by minimizing an upper bound on the expected loss function after $H$ local iterations. However, the practical quantity of interest is the validation accuracy. Given the non-convexity of the considered problem, decreasing the training loss as quickly as possible often leads to suboptimal solutions, especially in the case of supervised image classification. How do we reconcile the theory derived with respect to the training loss, and the practical quantity of interest, generalization and validation accuracy?
* Proposed adaptation strategy requires double the communication overhead of Synchronous SGD with gradient accumulation, however the authors are explicit about this limitation in the work.
* Numerical experiments are only conducted for supervised image classification with convolutional networks, however the authors are also explicit about this limitation in the work.
* According to details in the appendix, momentum is used in all numerical experiments, however it is not clear to me how this is incorporated in practice, especially in the case of Local SGD. I am concerned that local momentum buffers quickly fall out of synch, and this will exacerbate performance of the Local SGD baseline with heuristic linear or square-root scaling of the learning rate. Are there both global and local momentum buffers? If so, how do they interact? If not, how do you synchronize momentum buffers? Would properly handling momentum buffers change the conclusions or decrease the performance gap? For instance, there is Slow-Momentum [1] or BMUF [2], which take special care in handling the momentum buffers in distributed optimization.
* Based on the training loss curves in the appendix, it appears to me that all methods are still trained with a step-wise learning-rate decay. Is this indeed the case, that all numerical experiments are conducted with a step-wise learning rate decay? How does such a learning-rate schedule interact with the proposed Local AdaScale learning-rate scaling rule, and how do you reconcile this gap between theory and practice?
* Why specifically focus on an AdaScale type derivation of the scaling rule, especially since the choice to minimize an upper bound on the training loss in each iteration is somewhat arbitrary given that the relevant quantity at the focus of this work is generalization/validation accuracy? For instance, why not consider Adagrad or Adam style updates?


[1] Wang et al., Slowmo: Improving communication-efficient distributed sgd with slow momentum, ICLR 2020.

[2] Chen et al., Scalable training of deep learning machines by incremental block training with intra-block parallel optimization and blockwise model-update filtering, ICASSP 2016.

---

> ### Author Response · Authors · 2023-11-06
> **Addressing comments**
>
> Dear Reviewer yXqE,
>
> Thank you for your comments. We address your requested changes below.
>
> **Q1. Discrepancy between training loss vs validation performance.**
>
> A1. While purported effects of optimization methods on generalization are debated extensively in the community, we usually study optimization methods separately from the question of generalization. We chose to study LocalSGD purely as a distributed optimization method. We also want to highlight that AdaScale can be motivated purely from the optimization viewpoint. As we show in Section 5.1, AdaScale can be derived as a method that greedily maximizes the expected decrease in training loss for each step. We apply the same reasoning to LocalSGD.
>
> **Q2. Clarify whether the numerical experiments are conducted with a pre-specified step-wise learning rate decay rule on top of the proposed Local AdaScale rule**
>
> A2. We scale the base case’s ($K=1$, $H=1$) learning rate using the gain ratios computed in Equations 9 and 14. The learning rate schedule is considered part of the base case’s learning rate. In the case of adaptive learning rate methods, we replace the epoch counter that decides when to activate the learning rate scheduler with a `scale invariant’ epoch counter. This the $s$ variable in Algorithm 1 on Line 1 and 11.
>
> We use step decay only when the standard hyperparameters of the base case are so. This is true for the ImageNet and ImageNet32 experiments, but for the CIFAR-10 experiment we use a cosine annealing (see Appendix H.1 for hyperparameters), and the smoother gain ratio curves for CIFAR-10 in Figure 12 (Appendix) compared to Figure 13 which uses step decay. This is not a discrepancy between theory and practice.
>
>
> **Q3. How are momentum buffers handled?**
>
> A3. We follow the original AdaScale paper who argue that momentum is of secondary importance to performance when using momentum SGD, and do not tune it across various $K$. In practice, each worker maintains an independent momentum buffer that is not synchronized. Synchronizing it would add significantly to the communication overhead.
>
> We limit ourselves to simple averaging of the weights and defer the analysis of Slow-mo-like averaging to future work.
>
> **Q4. Paper on learning rate adaptation**
>
> We will try to broaden the discussion on learning rate adaptation techniques in the related work section of the final draft of the paper.

---

### Review · Reviewer_25QM · 2023-10-31

**Summary Of Contributions:**

The standard distributed data-parallel training of neural nets requires constant synchronization of the gradients between workers. Using Local SGD is one way to reduce the communication overhead, but its gradient estimate is biased. This paper borrows the idea of AdaScale to study how the learning rate should be set automatically for Local SGD. More specifically, their method adjusts the learning rates according to the gradient statistics and may automatically add more training steps until loss converges.

By comparing the empirical performances of SGD and Local SGD, the authors claim that Local SGD is empirically preferable to SGD only in extreme scenarios of communication being substantially more time-consuming than computation.

**Audience:**

Yes

**Broader Impact Concerns:**

This work studies the optimization methods in deep learning and raises no ethical issues.

**Claims And Evidence:**

Yes

**Requested Changes:**

The paper is of high quality overall and I would like to vote for acceptance. However, I encourage the authors to rephrase their main claim on the practical utility of Local SGD and make a clearer distinction between optimization and generalization.

Typos:
* Page 3, Contribution 3: "Local SGD convergences faster" -> converges
* Page 12, "our findings on the value of Local SGD might limited to those conditions." -> might be

**Strengths And Weaknesses:**

Strengths:
1. This paper studies a well-motivated problem: how do we reduce communication overhead while maintaining the convergence speed? The authors tackle this problem by looking into the learning rates of Local SGD, one of the most popular communication-efficient optimizers.
1. This paper is well-written. The experiment and proof details are written very clearly.
2. The proposed method LocalAdaScale is grounded by theory and has been validated via thorough experiments on different datasets, numbers of local steps, and numbers of workers.

Weaknesses:
1. My main concern is about the main claim of the paper, "Local SGD is of limited practical utility". In my understanding, what the authors indeed did is to design learning rate schedules with their best efforts (via LocalAdaScale), and found that Local SGD is not faster than SGD if the communication cost is not strikingly large. I believe a more fair summary of their efforts could be "tuning learning rates of Local SGD alone may not be able to gain much over SGD" because other hyperparameters may also be very important, but this paper doesn't include any empirical study on them. For example, although this paper mainly views Lin et al. (2020) as a paper showing negative results of Local SGD, what they actually do in the paper is to propose a method, called "Post-local SGD", which is Local SGD with a special schedule of $H$: it starts with 1 and then increases to a larger constant halfway through training. In fact, Lin et al. (2020) found that this method can even beat SGD with the same number of training steps. So tuning the schedule of H, instead of tuning the schedule of learning rates, may improve the practical utility of Local SGD, but this paper doesn't have any related experiments. I would recommend the authors restrict their claim to tuning the learning rates alone.
2. Another issue of this paper is that it doesn't make a clear distinction between the previous efforts studying optimization and generalization.
    * Many tricks on learning rates, such as learning rate decay and warmup, are not only aiming for faster optimization. A folklore observation in deep learning is that starting training with a small learning rate can be much faster than that with a large learning rate, but in the end, the test accuracy is worse. The theory part of this paper only studies the learning rate from the optimization aspect, but in the experiments, the test accuracy is always reported. It is thus unclear whether the learning rates found by AdaScale are indeed "optimal" for test accuracy.
    * Previous works such as Lin et al. (2020) and Gu et al. (2023) are studying the generalization aspect of Local SGD. Their claim is that Local SGD (with LR and H set properly) can achieve better test accuracy, although the training loss could be a bit worse than SGD. All their experiments are conducted in standard setups with CIFAR-10 and ImageNet, so it is unfair to ignore the results and claim the scenarios being studied are "impractical" (Page 4). I would recommend the authors write more clearly that the current paper only studies Local SGD in terms of the convergence speed of training loss, and report the training loss in all their figures.

---

> ### Author Response · Authors · 2023-11-06
> **Addressing comments**
>
> We thank the reviewer for the detailed comments.
>
> **Restriction to learning rate tuning**
>
> You are right to point out that our statement “LocalSGD is of limited practical utility” has to be put into the right context. We will clarify our focus on learning rate tuning in the upcoming revision and make clear that other hyperparameters might have an important role to play as well.
>
> In particular, as you rightfully point out, $H$ may be a crucial hyperparameter in practice to balance performance and communication overhead. However, we want to point out that, at any given “schedule” for $H$, we could likewise do a comparison between LocalSGD and gradient accumulation under linear and adaptive scaling. We believe that it is adequate for this paper to restrict the comparison to a fixed $H$.
>
> That said, the considerations we use to derive LocalAdaScale may likewise be useful to inform the setting of $H$, as we hint in the Conclusion section. We leave this to future work.
>
> **Impractical comment on Gu et al. (2023)**
>
> The claim by Gu et al is that Local SGD generalizes better than SGD when using “small enough learning rate and training and long enough”. The tools proposed in that work are undoubtedly great for analysis, in practice we are interested in getting to a performance in the shortest time possible.  Our impractical comment was purely in regard to this. We will clarify this in the paper.
>
> **Generalization vs optimization**
>
> While purported effects of optimization methods on generalization are debated extensively in the community, we usually study optimization methods separately from the question of generalization. We chose to study LocalSGD purely as a distributed optimization method. We also want to highlight that AdaScale can be motivated purely from the optimization viewpoint. As we show in Section 5.1, AdaScale can be derived as a method that greedily maximizes the expected decrease in training loss for each step. We apply the same reasoning to LocalSGD.

---

### Author Response · Authors · 2023-11-11
**Updated PDF**

Dear Reviewers,

Thank you for your detailed reviews.

We have uploaded a newer version that addresses your comments. The major changes are:

* Emphasising that the main claims are based on learning rate scaling -- Reviewer 25QM
* Differentiating optimization and generalization. -- Reviewers 25QM, yXqE
* Adding details clarifying the usage of iterations and epochs in experiments -- Reviewer 3XYf

All changes have been made in coloured text.

We hope these changes sufficiently address your any remaining questions.

---

### Public Comment · ~Ali3 · 2024-09-01
**Bunch of Amateur Questions**

It was a very attractive work. However, I have some blurry points as an amateur.

1. LR also updates after each H steps?
2. I implemented your approach step by step, but after each H step while calculating the gain ratio, because unreasonable value of G bar (which is surprisingly negative sometimes) according the algorithm 15, I reach to some negative values for gain ratio (and high values for LR(like 0.7 or even 1.1) if we consider that the answer of the question 1 is positive.).

Thank you in advance

---

### Decision · Action_Editor_WerS · 2023-12-21

**Recommendation:** Accept as is

**Comment:**

All three reviewers also recommend this be accepted, in particular after a slight adjustment in the claims. I agree with them.

**Audience:**

The paper carefully looks into the question if by choosing an adaptive learning rate, can local SGD be faster/useful. I think the distributed learning/optimization community will find this interesting.

**Claims And Evidence:**

The paper consider the role of learning rates on local SGD, how it affects the bias in the gradient updates, and overall convergence. It then suggests an adaptive learning rate (similar to Adascale) that brings the convergence of Local SGD more in line with that of mini-batching and SGD. Overall, even with the adaptive learning rate, there is little gain to be had in using local SGD unless there is a very elevated communication cost. I find the paper to be clearly written, and of interest to those in distributed learning, and thus recommend it be accepted.

---

> ### Author Response · Authors · 2024-01-13
> **Thank you & Updates on the camera ready**
>
> Dear Editor,
>
> Thank you for the feedback and the recommendation for acceptance. We have incorporated the additional change about the learning rate reparametrization, and have uploaded the camera ready.
>
> Thank you, again.
>
> Authors